# Reinforced Sequential Monte Carlo for Amortised Sampling

**Sanghyeok Choi** [1 2 3]  **Sarthak Mittal** [3 4]  **Víctor Elvira** [1]  **Jinkyoo Park** [2]  **Esmeralda S. Whitammer** [1 5]

## Abstract

This paper proposes a synergy of amortised and particle-based methods for sampling from distributions defined by unnormalised density functions. We state a connection between sequential Monte Carlo (SMC) and neural sequential samplers trained by maximum-entropy reinforcement learning (MaxEnt RL), wherein learnt sampling policies and value functions define proposal kernels and twist functions. Exploiting this connection, we introduce an off-policy RL training procedure for the sampler that uses samples from SMC – using the learnt sampler as a proposal – as a behaviour policy that better explores the target distribution. We describe techniques for stable joint training of proposals and twist functions and an adaptive weight tempering scheme to reduce training signal variance. Furthermore, building upon past attempts to use experience replay to guide the training of neural samplers, we derive a way to combine historical samples with annealed importance sampling weights within a replay buffer. On synthetic multi-modal targets (in both continuous and discrete spaces) and the Boltzmann distribution of alanine dipeptide conformations, we demonstrate improvements in approximating the true distribution as well as training stability compared to both amortised and Monte Carlo methods. Code is available at `https://github.com/hyeok9855/ReinforcedSMC`.

## 1. Introduction

The problem of sampling from high-dimensional unnormalised probability distributions arises in many domains, including Bayesian inference of statistical model parameters

[1]University of Edinburgh [2]KAIST [3]Mila – Québec AI Institute [4]Université de Montréal [5]CIFAR Fellow. Correspondence to: Sanghyeok Choi <sanghyeok.choi@ed.ac.uk>.

*Proceedings of the 43rd International Conference on Machine Learning*, Seoul, South Korea. PMLR 306, 2026. Copyright 2026 by the author(s).

(Welling & Teh, 2011) and sampling of molecular conformations in computational chemistry (Noé et al., 2019; Holdijk et al., 2023). Solving this problem requires exploration of a complex, multi-modal energy landscape defining a Boltzmann distribution that is intractable to sample from directly. Classical approaches such as Monte Carlo (MC) methods have been used to address this challenge for decades. These approaches include Markov chain Monte Carlo (MCMC) methods like Hamiltonian Monte Carlo (HMC; Duane et al., 1987; Hoffman et al., 2014) and accelerated Langevin dynamics (Leimkuhler et al., 2014), importance sampling algorithms and their adaptive variants (Bugallo et al., 2017), and particle-based methods like sequential Monte Carlo (SMC; Halton, 1962; Del Moral et al., 2006).

Recent work has considered amortised methods that train (hierarchical) latent variable models (Ranganath et al., 2016) parametrised with neural networks to directly sample from the target distribution. Examples of such models include autoregressive sequence models and diffusion models (Ho et al., 2020). These families can capture complex distributions by composing sequences of simple conditional distributions. Diffusion models, for example, have shown promise for amortised sampling: in contrast to diffusion models trained to maximise a variational bound on log-likelihood of a dataset (Ho et al., 2020; Song et al., 2021b;a), these *diffusion samplers* are trained, by one of many available objectives linked with stochastic optimal control, assuming access to a queryable energy function (*e.g.*, Zhang & Chen (2022); Vargas et al. (2023), see related work in §A). Methods for amortised sampling in discrete spaces, where connections to reinforcement learning (RL) are more explicit, have also been proposed (Bengio et al., 2021).

In contrast to MC methods without learnable components, amortised samplers with latent variable models can take advantage of the generalisation abilities of deep networks to exploit regularities in the target distribution. In addition, while MC methods typically require a large number of steps or samples to produce low-variance estimators in challenging high-dimensional scenarios (Robert et al., 1999; Liu & Liu, 2001; Owen, 2013), amortised samplers can generate samples in finite time once trained. On the other hand, the anytime nature of MC methods can be seen as a strength: MC methods are designed to approach the target distribution as the number of steps or particles increases, while amor-

tised samplers may fail to converge to the target distribution due to insufficient model capacity or optimisation issues such as mode collapse.

In this work, we propose a scheme for training amortised hierarchical samplers that combines the strengths of both amortised and MC methods, wherein the MC method benefits from the learnt components of the amortised sampler, while the amortised sampler is trained using off-policy samples from the MC method. The paper is structured as follows:

**Unifying HVI, MaxEnt RL, and SMC/AIS (§2).** We give a unified account of hierarchical variational inference (HVI), maximum-entropy reinforcement learning (MaxEnt RL) for sampling, and sequential Monte Carlo (SMC) with annealed importance sampling (AIS), aiming to elucidate the mathematical connections between these three areas. This is the first, to our knowledge, treatment of these three topics in a single framework, and it sets the stage for our proposed algorithms.

**A synergy of Monte Carlo and amortised sampling (§3).** We describe an efficient training procedure for amortised sequential samplers that uses samples from SMC as a behaviour policy for off-policy RL training. We describe design considerations and stabilisation techniques for training of the proposal kernels and twist functions used in SMC, an adaptive weight tempering scheme that reduces variance during training, and a principled way to combine weighted training samples with prioritised experience replay, a commonly used method in RL.

**Experiments (§4).** We demonstrate the effectiveness of our method on multi-modal target distributions in both continuous and discrete spaces, including the Boltzmann distribution of a common benchmark in molecular simulation. Our approach consistently improves target distribution approximation and mode coverage compared to methods from prior work.

## 2. Amortised and Particle Methods for Sampling

We address the problem of sampling from a target probability distribution $\pi$ on a space $\mathcal{X}$, which will be either $\mathbb{R}^d$ or a discrete space. The target distribution takes the form $\pi(\mathbf{x}) = R(\mathbf{x})/Z$ where the unnormalised density $R : \mathcal{X} \to \mathbb{R}_{>0}$ can be queried pointwise but $Z = \int_{\mathcal{X}} R(\mathbf{x})\,\mathrm{d}\mathbf{x}$ is unknown.[1]

---

[1]All distributions are assumed to have full support and be absolutely continuous with respect to the counting or Lebesgue measure, and we can thus abuse notation and use them interchangeably with their positive density or mass functions. In the discrete case, integrals below should be interpreted w.r.t. the counting measure, *i.e.*, as sums.

### 2.1. Amortised Sampling with Hierarchical Latent Variable Models

In hierarchical variational inference (Ranganath et al., 2016), one assumes a latent variable model $\overrightarrow{p}_\theta$ with a Markov chain structure:

$$\mathbf{x}_0 \xrightarrow{\overrightarrow{p}_\theta} \ldots \xrightarrow{\overrightarrow{p}_\theta} \mathbf{x}_{n-1} \xrightarrow{\overrightarrow{p}_\theta} \mathbf{x}_N = \mathbf{x}, \qquad (1)$$

where the intermediate variables $\mathbf{x}_0, \ldots, \mathbf{x}_{N-1}$ lie in some spaces $\mathcal{X}_n$ that may be either copies of $\mathcal{X}$ or some auxiliary spaces. The initial distribution $\overrightarrow{p}_0(\mathbf{x}_0)$ and conditional distributions $\overrightarrow{p}_\theta(\mathbf{x}_{n+1} \mid \mathbf{x}_n)$, which depend on $n$ and on a model parameter $\theta$, define a joint distribution over $\mathbf{x}_{0:N} = (\mathbf{x}_0, \mathbf{x}_1, \ldots, \mathbf{x}_N)$:

$$\overrightarrow{p}_\theta(\mathbf{x}_{0:N}) = \overrightarrow{p}_0(\mathbf{x}_0) \prod_{n=0}^{N-1} \overrightarrow{p}_\theta(\mathbf{x}_{n+1} \mid \mathbf{x}_n) \qquad (2)$$

The goal of amortised sampling is to fit $\theta$ so as to make the marginal distribution over $\mathbf{x}$,

$$\overrightarrow{p}_\theta(\mathbf{x}) = \int \overrightarrow{p}_\theta(\mathbf{x}_{0:N})\,\mathrm{d}\mathbf{x}_0\,\mathrm{d}\mathbf{x}_1 \ldots \mathrm{d}\mathbf{x}_{N-1}, \qquad (3)$$

close to the target $\pi(\mathbf{x})$ in some measure of divergence. Once trained, it requires only one fixed-time rollout to produce samples, amortising the cost of a runtime sampler (MC methods) into a parametrised model.

The intractability of the integral in (3) motivates introducing a posterior distribution with factorisation in the reverse order:

$$\overleftarrow{p}(\underbrace{\mathbf{x}_0, \ldots, \mathbf{x}_{N-1}}_{\mathbf{x}_{0:N-1}} \mid \mathbf{x}) = \prod_{n=0}^{N-1} \overleftarrow{p}(\mathbf{x}_n \mid \mathbf{x}_{n+1}). \qquad (4)$$

One then minimises a divergence between the joint distributions $\overrightarrow{p}_\theta(\mathbf{x}_{0:N})$ and $\pi(\mathbf{x})\overleftarrow{p}(\mathbf{x}_{0:N-1} \mid \mathbf{x})$, which (for suitable divergences) upper-bounds the divergence between the marginals $\overrightarrow{p}_\theta(\mathbf{x})$ and $\pi(\mathbf{x})$ by the data processing inequality. A typical choice is the reverse Kullback-Leibler (KL) divergence $\mathrm{KL}(\overrightarrow{p}_\theta(\mathbf{x}_{0:N}) \,\|\, \pi(\mathbf{x})\overleftarrow{p}(\mathbf{x}_{0:N-1} \mid \mathbf{x}))$. Note that in general the reverse transition kernel can also have learnable parameters (*e.g.*, Richter et al. (2024); Gritsaev et al. (2025); Blessing et al. (2025) in the case of diffusion samplers, where the reverse kernel defines the noising process), but here we limit our discussion to fixed reverse kernels.

This formulation of latent variable models is sufficiently general to apply to variables defined in both continuous and discrete spaces. We demonstrate this flexibility through two example models used in our experiments (§4): diffusion models for continuous variable generation and prepend/append models for discrete sequence generation. Details are given in §B.

## 2.2. Off-Policy Entropic RL for HVI

Here we state the connection between amortised sampling and off-policy RL methods; see Tiapkin et al. (2024); Deleu et al. (2024) for expanded expositions.

The problem of matching the distribution $\overrightarrow{p}_\theta(\mathbf{x}_{0:N})$ to $\pi(\mathbf{x})\overleftarrow{p}(\mathbf{x}_{0:N-1} \mid \mathbf{x})$ – where the latter is not tractable to sample from and known only up to normalising constant $Z$ – can be cast as one of maximum-entropy reinforcement learning (Haarnoja et al., 2018) in a deterministic Markov decision process (MDP). In this formulation, states are pairs $(n, x_n)$, where $n \in \{0, 1, \ldots, N\}$ is an index and $x_n$ is a value assigned to the variable $\mathbf{x}_n$, augmented by a special terminal state $\top$. The actions are values assigned to the next variable $\mathbf{x}_{n+1}$, and the transition dynamics are deterministic: taking action $x_{n+1}$ from state $(n, x_n)$ leads to state $(n + 1, x_{n+1})$. The initial state distribution over states $(0, x_0)$ is given by $\overrightarrow{p}_0(\mathbf{x}_0)$, and the episode arrives after $N$ steps in a terminal state $(N, x)$, after which transition to $\top$ is forced. Policies in this environment are equivalent to the conditional distributions $\overrightarrow{p}_\theta(\mathbf{x}_{n+1} \mid \mathbf{x}_n)$, and the joint distribution over the variables, $\overrightarrow{p}_\theta(\mathbf{x}_{0:N})$, is equivalent to the distribution over episodes induced by the policy. If one appropriately defines the stepwise rewards as

$$r((n, x_n), x_{n+1}) = \log \overleftarrow{p}(\mathbf{x}_n \mid \mathbf{x}_{n+1}), \quad n < N,$$
$$r((N, x), \top) = \log R(\mathbf{x}), \tag{5}$$

then the sum of rewards along the trajectory $x_{0:N+1}$ is equal to $\log(R(\mathbf{x})\overleftarrow{p}(\mathbf{x}_{0:N-1} \mid \mathbf{x}))$.

The objective of maximum-entropy RL is to learn a policy that maximises the expected sum of rewards and entropies of the policies along the trajectory:

$$\max_\theta \mathbb{E}_{x_{0:N+1} \sim \overrightarrow{p}_\theta} \left[ \sum_{n=0}^{N-1} r((n, x_n), x_{n+1}) + \mathcal{H}[\overrightarrow{p}_\theta(\cdot \mid (n, x_n))] \right.$$
$$\left. + r((N, x), \top) \right]. \tag{6}$$

It can be shown (see, *e.g.*, Eysenbach & Levine (2022)) that this objective is equivalent to minimising the KL divergence $\mathrm{KL}(\overrightarrow{p}_\theta(\mathbf{x}_{0:N}) \| \pi(\mathbf{x})\overleftarrow{p}(\mathbf{x}_{0:N-1} \mid \mathbf{x}))$. It follows that the optimal policy is the one that solves the amortised inference problem.

**Objectives.** Various objectives exist for training the policy to solve the problem (6), all of which at optimality enforce the soft Bellman equations (Rust, 1987; Haarnoja et al., 2018) for all states. In the sampling setting described here, these objectives are most concisely stated in probabilistic formulations as originally proposed in the generative flow network (GFlowNet) framework (Bengio et al., 2021; 2023; Lahlou et al., 2023). We next state two such objectives, trajectory balance and subtrajectory balance (Malkin et al.,

2022; Madan et al., 2023). As shown by Deleu et al. (2024), both amount to cases of *path consistency learning* in entropic RL (Nachum et al., 2017). In the following sections, we exclusively use $\mathbf{x}_n$ notation (not $x_n$), unless a distinction is required.

In the notations of this paper, we model *flow functions* $F_n^\phi : \mathcal{X}_n \to \mathbb{R}_{>0}$, $n = 0, \ldots, N - 1$, parametrised by $\phi$, and set $F_N^\phi(\mathbf{x}) = R(\mathbf{x})$. These flow functions correspond to exponentiated soft value functions in the wider RL literature. These connections extend to reward-based fine-tuning of diffusion models, where similar properties have been observed (Uehara et al., 2024; Venkatraman et al., 2024; Deleu et al., 2025).

For a trajectory $\mathbf{x}_{0:N} = (\mathbf{x}_0, \ldots, \mathbf{x}_N = \mathbf{x})$, and every subtrajectory $\mathbf{x}_{m:n} = (\mathbf{x}_m, \ldots, \mathbf{x}_n), 0 \le m < n \le N$, the subtrajectory balance (SubTB) loss is defined as

$$\mathcal{L}_{\mathrm{SubTB}}^{\theta,\phi}(\mathbf{x}_{m:n}) = \left[ \log \frac{F_m^\phi(\mathbf{x}_m) \prod_{i=m}^{n-1} \overrightarrow{p}_\theta(\mathbf{x}_{i+1} \mid \mathbf{x}_i)}{F_n^\phi(\mathbf{x}_n) \prod_{i=m}^{n-1} \overleftarrow{p}(\mathbf{x}_i \mid \mathbf{x}_{i+1})} \right]^2. \tag{7}$$

The trajectory balance (TB) loss (Malkin et al., 2022) is the special case of SubTB for $m = 0$ and $n = N$:

$$\mathcal{L}_{\mathrm{TB}}^{\theta,\phi}(\mathbf{x}_{0:N}) = \left[ \log \frac{F_0^\phi(\mathbf{x}_0) \overrightarrow{p}_\theta(\mathbf{x}_{1:N} \mid \mathbf{x}_0)}{R(\mathbf{x}_n) \overleftarrow{p}(\mathbf{x}_{0:N-1} \mid \mathbf{x}_n)} \right]^2. \tag{8}$$

If $\overrightarrow{p}_0(\mathbf{x}_0)$ is fixed, one can set $F_0^\phi(\mathbf{x}_0) = Z_\theta \cdot \overrightarrow{p}_0(\mathbf{x}_0)$, where $Z_\theta$ is a learnable scalar parameter, eliminating $\phi$ from the TB loss.

The TB loss (8) alone, enforced on *all* trajectories $\mathbf{x}_{0:N}$, is sufficient to ensure that a policy $\overrightarrow{p}_\theta$ satisfies $Z_\theta \overrightarrow{p}_\theta(\mathbf{x}_{0:N}) = R(\mathbf{x})\overleftarrow{p}(\mathbf{x}_{0:N-1} \mid \mathbf{x})$ at optimality, in which case $Z_\theta$ equals the true normalising constant $Z$ of the distribution. Notice that TB does not involve any intermediate flow functions $F_n^\phi$ for $n < N$.

The SubTB loss (7), on the other hand, involves the intermediate flow functions $F_n^\phi$ and can be enforced on any subtrajectories $\mathbf{x}_{m:n}$. SubTB is a stronger condition than TB alone, and it has been argued that this leads to better credit assignment and training stability (Madan et al., 2023).[2] In practice, one may choose to enforce SubTB on a subset of subtrajectories; we use the scheme described in §C. At optimality, if SubTB is minimised to 0 for all $\mathbf{x}_{0:N}$, the intermediate flows $F_n^\phi$ are proportional to the marginal distributions $\overrightarrow{p}_\theta(\mathbf{x}_n)$.

---

[2]Subtrajectory objectives were studied for diffusion samplers in Zhang et al. (2024) but found in Sendera et al. (2024) to underperform TB and its close relative, log-variance loss (Richter et al., 2020), due to instability. However, results in the few-step setting (Berner et al., 2026) suggest SubTB may see its intended credit assignment benefits realised with appropriate behaviour policies such as the ones we propose here.

**Off-policy training.** A unique advantage of objectives such as SubTB and TB is that they can be optimised, by gradient updates w.r.t. parameters $\theta$ and $\phi$, with *off-policy* trajectories, *i.e.*, ones sampled from an arbitrary full-support *behaviour policy*, without requiring importance sampling corrections. This contrasts with hierarchical variational inference methods that directly optimise the reverse KL on full or partial trajectories (Ranganath et al., 2016; Sobolev & Vetrov, 2019; Zimmermann et al., 2023), which suffer from high gradient variance when using off-policy sampling due to importance weighting (Malkin et al., 2023). This enables more effective exploration of multi-modal target distributions through advanced exploratory behaviour policies (Sendera et al., 2024; Kim et al., 2025a;b).

### 2.3. SMC and AIS

Sequential Monte Carlo samplers (Del Moral et al., 2006) are a class of particle-based algorithms that approximate a target distribution by propagating a set of latent particles through a sequence of intermediate distributions, yielding a collection of weighted samples that is an empirical approximation to the target. Here we state SMC samplers in the notations introduced above.

We fix a sequence of *intermediate target distributions* $\pi_n(\mathbf{x}_n) = F_n(\mathbf{x}_n)/Z_n$ on the spaces $\mathcal{X}_n$, $n = 1, \ldots, N$, with $F_N = R$ (so $\pi_N = \pi$), given by unnormalised densities $F_n : \mathcal{X}_n \to \mathbb{R}_{>0}$, and set $F_0 = \pi_0 = \overrightarrow{p}_0$. We also assume a pair of *proposal kernels*, suggestively denoted $\overrightarrow{p}(\mathbf{x}_{n+1} \mid \mathbf{x}_n)$ and $\overleftarrow{p}(\mathbf{x}_n \mid \mathbf{x}_{n+1})$, that define a Markov chain on the spaces $\mathcal{X}_n$ in the forward and backward directions, respectively.

The basic SMC sampler proceeds as follows:

(1) Initialise i.i.d. particles $\mathbf{x}_0^k \sim \pi_0(\mathbf{x}_0)$ with uniform weights $w_0^k = 1/K$ for $k = 1, \ldots, K$.
(2) For $n = 0, \ldots, N-1$:
 (a) Propose new particles $\mathbf{x}_{n+1}^k \sim \overrightarrow{p}(\mathbf{x}_{n+1} \mid \mathbf{x}_n^k)$.
 (b) Update weights:

$$w_{n+1}^k = w_n^k \cdot \overbrace{\frac{F_{n+1}(\mathbf{x}_{n+1}^k) \overleftarrow{p}(\mathbf{x}_n^k \mid \mathbf{x}_{n+1}^k)}{F_n(\mathbf{x}_n^k) \overrightarrow{p}(\mathbf{x}_{n+1}^k \mid \mathbf{x}_n^k)}}^{=: \tilde{w}_n^k} . \quad (9)$$

 (c) Optionally, resample the particles in proportion to their weights to obtain an unweighted set of particles (or variations thereof).
(3) Return the weighted particles $\{(\mathbf{x}_N^k, w_N^k)\}_{k=1}^K$.

The resulting approximation $\sum_{k=1}^K W_N^k \delta_{\mathbf{x}_N^k}(\mathbf{x})$, where $W_N^k = w_N^k / \sum_{j=1}^K w_N^j$, is an empirical approximation that converges weakly to $\pi(\mathbf{x})$ as $K \to \infty$ under mild conditions (Del Moral et al., 2006). The intermediate particles and weights similarly approximate the intermediate targets $\pi_n(\mathbf{x}_n)$.

In the case where the resampling step is omitted, a telescoping cancellation occurs in the weights accumulated by (9), and the final weights simplify to

$$w_N^k = \frac{R(\mathbf{x}_N^k) \prod_{n=0}^{N-1} \overleftarrow{p}(\mathbf{x}_n^k \mid \mathbf{x}_{n+1}^k)}{\overrightarrow{p}_0(\mathbf{x}_0^k) \prod_{n=0}^{N-1} \overrightarrow{p}(\mathbf{x}_{n+1}^k \mid \mathbf{x}_n^k)}. \quad (10)$$

In this case, the algorithm is known as annealed importance sampling (AIS; Neal, 2001), which is simply importance weighting on the level of trajectories $\mathbf{x}_{0:N}$ with proposal $\overrightarrow{p}_0(\mathbf{x}_0) \overrightarrow{p}(\mathbf{x}_{1:N} \mid \mathbf{x}_0)$ and target $R(\mathbf{x}) \overleftarrow{p}(\mathbf{x}_{0:N-1} \mid \mathbf{x})$.

The performance of SMC crucially depends on the choices of the intermediate targets $\pi_n$ and the proposal kernels $\overrightarrow{p}$ and $\overleftarrow{p}$. A common choice for the targets is to use geometric annealing:

$$F_n(\mathbf{x}) = \overrightarrow{p}_0(\mathbf{x})^{1-\beta_n} R(\mathbf{x})^{\beta_n}, \quad (11)$$

where $0 = \beta_0 < \beta_1 < \cdots < \beta_N = 1$ is a sequence of inverse temperatures. The proposal kernels are often chosen to be MCMC kernels to which the intermediate targets are invariant (*e.g.*, Langevin dynamics on $F_n$). *Adaptive* importance sampling methods (Cappé et al., 2004; Cornuet et al., 2012; Bugallo et al., 2017; Elvira & Chouzenoux, 2022, *inter alia*) refine the proposal kernels based on the current population of particles. This work can be seen as a further step in this direction, where the proposal kernels are learnt through off-policy RL, as we describe in the next section.

## 3. Methods

### 3.1. Policies and Flows as Proposals and Twisted Targets

Notice that the logarithm of the importance weight (10) matches the expression inside the square in the TB objective (8) evaluated on the trajectory $\mathbf{x}_{0:N}$, and TB averaged over a batch of trajectories equals second moment of the log-AIS weights on those trajectories about $\log Z_\theta$.[3] Similarly, the mean SubTB objective (7) equals the second moment of the log-weights accumulated between steps $m$ and $n$. In particular, if SubTB is minimised to 0 on subtrajectories of length 1, then detailed balance is satisfied between the intermediate targets $\pi_n$ and the proposal kernels $\overrightarrow{p}$ and $\overleftarrow{p}$, *i.e.*, $\pi_n(\mathbf{x}_n) \overrightarrow{p}(\mathbf{x}_{n+1} \mid \mathbf{x}_n) = \pi_{n+1}(\mathbf{x}_{n+1}) \overleftarrow{p}(\mathbf{x}_n \mid \mathbf{x}_{n+1})$ for all $n$, in which case the weights $w_n^k$ remain uniform at all steps and resampling is not required.

**These observations motivate the use of amortised samplers, trained by the methods in §2.2, within the SMC algorithm described in §2.3.** Given an amortised sampler trained by TB or SubTB, with policies $\overrightarrow{p}_\theta$ and flow

---

[3]Cf. the VarGrad objective (Richter et al., 2020), which optimises the *variance* of log-importance weights and is independent of $\log Z_\theta$, and empirically has very similar behaviour to TB for diffusion sampling (Sendera et al., 2024).

functions $F_n^\phi$, one can perform SMC using the policies $\overrightarrow{p}_\theta$ as the proposals $\overrightarrow{p}$ and using the flows $F_n^\phi$ as the unnormalised intermediate target densities $F_n$. Optimality of the sampler with respect to the SubTB loss implies optimality of SMC with the corresponding proposals and intermediate targets. If only the TB loss is used, so flows are not trained, vanishing of the loss implies optimality of AIS. In both cases, optimality implies convergence to the same policies and uniform importance weights – whether trajectory- or subtrajectory-level.

We empirically find it best to update the policy/proposal parameters $\theta$ using *only* the TB loss and the flow/target parameters $\phi$ using *only* the SubTB loss. Other combinations are suboptimal or unstable; see §G.2. We note that the use of an imperfectly trained diffusion sampler as an SMC proposal has concurrently been proposed by Wu et al. (2025), who nonetheless do not consider training the sampler with off-policy samples from SMC, as we discuss next.

### 3.2. SMC as Behaviour Policy

While the amortised sampler provides proposal kernels for SMC (§3.1), we propose to use samples obtained by SMC to train the amortised sampler, creating a mutually beneficial cycle. Because SMC produces samples that approximate the target distribution and can explore regions of the target space that have high probability under $\pi$ but low probability under the current sampler $\overrightarrow{p}_\theta$, using these samples can guide the training of the sampling policy more efficiently than on-policy samples. This proposition is in line with other methods of obtaining approximate target samples to guide off-policy RL training, *e.g.*, Langevin dynamics (Sendera et al., 2024; Phillips & Cipcigan, 2024), partial destruction and regeneration of known samples (Zhang et al., 2022; Hu et al., 2023), or metaheuristic methods (Kim et al., 2024a; 2025a).

The **simplest form** of our proposed method takes the behaviour policy to be a mixture of two distributions over trajectories:

- The current sampler $\overrightarrow{p}_\theta$ (on-policy component);
- Trajectories sampled from $\overleftarrow{p}(\mathbf{x}_{1:N-1} \mid \mathbf{x}_N)$, where $\mathbf{x}_N$ are samples obtained from SMC with proposals $\overrightarrow{p}_\theta$ and intermediate targets defined by the current flows $F_n^\phi$ (off-policy component).

This algorithm is presented as Algorithm 1 (AIS variant not using intermediate flows) and Algorithm 2 (SMC variant). As we will describe in the following sections, we consider more sophisticated variants that use past samples stored in a replay buffer and reweight them to better approximate the target distribution.

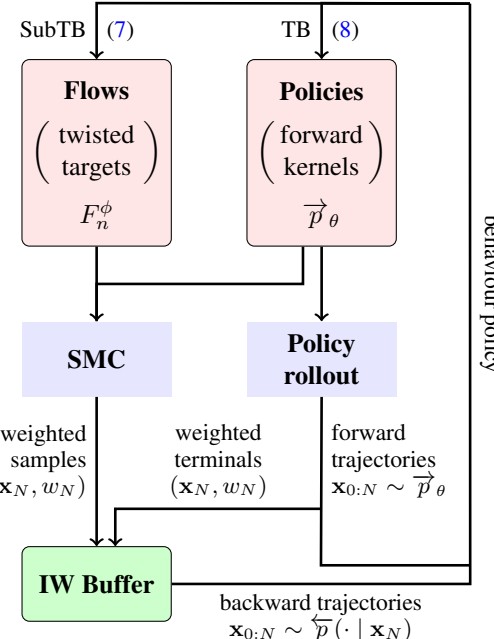

*Figure 1.* Objects and updates in reinforced sequential Monte Carlo training. The flows (intermediate targets) and policies (transition kernels) are trained using the TB (8) or SubTB losses (7) on trajectories sampled either by on-policy sampling (forward trajectories from $\overrightarrow{p}$) or off-policy sampling (backward trajectories from the importance-weighted buffer and $\overleftarrow{p}$).

### 3.3. Importance-Weighted Experience Replay

Experience replay is a common technique in off-policy RL that learns a policy with episodes generated during past iterations using a replay buffer. One of the key design choices for the replay buffer is how to combine historical samples, each generated with a different proposal distribution. Prioritised experience replay (Schaul et al., 2016) used the value of the loss function (temporal difference) as weights for samples in the buffer.

Building on the connection between MaxEnt RL and SMC/AIS that we have already established, we propose using importance weights for prioritisation. Consider a (prioritised) replay buffer $\mathcal{B}$ that stores $M$ batches, with each batch indexed by $m$ containing $K$ samples from $\overrightarrow{p}_{\theta_m}$ and $F_n^{\phi_m}$. To *properly* weight samples from different proposals, we first define batch-level weights, inspired by Martino et al. (2018a) (see their §B for details). The weight for a given batch is defined by its particle estimate of the normalising constant, $\widehat{Z}^m$, which depends on the sampling strategy used:

- For on-policy batches sampled with $\overrightarrow{p}_\theta$:

$$\widehat{Z}^m = \frac{1}{K} \sum_{k=1}^{K} w_N^{m,k}. \qquad (12)$$

- For off-policy batches sampled using SMC with resam-

pling performed at $n_r$ steps indexed by $0 = r_0 < r_1 < \cdots < r_{n_r} = N$:

$$\widehat{Z}^m = \prod_{j=1}^{n_r} \left( \sum_{k=1}^{K} W_{r_{j-1}}^{m,k} \prod_{i=r_{j-1}}^{r_j - 1} \widetilde{w}_i^{m,k} \right), \quad (13)$$

where $W_n^k = w_n^k / \sum_{j=1}^{K} w_n^j$ are the self-normalised weights at step $n$ and $\widetilde{w}_i^k$ is the incremental importance weights at step $i$, as defined in (9).

The final weight of an individual sample $\mathbf{x}^{m,k}$ is then defined by the product of batch-level weight $\widehat{Z}^m$ and the self-normalised weight within a batch $W_N^{m,k} = w^{m,k} / \sum_{j=1}^{K} w^{m,j}$, *i.e.*, the buffer can be written as $\mathcal{B} = \{(\mathbf{x}^{m,k}, \widehat{Z}^m W_N^{m,k})\}_{k=1,...,K}^{m=1,...,M}$. The weighted samples in the buffer themselves are also a particle approximation that converges weakly to $\pi(\mathbf{x})$ as $MK \to \infty$.

To obtain trajectories for training, we sample $\mathbf{x}$ with replacement from the buffer with probabilities proportional to their weights, and then sample backward trajectories using $\overleftarrow{p}(\mathbf{x}_{1:N-1} \mid \mathbf{x})$. In this way, the algorithm extends the basic procedure described in §3.2 so as to use past samples instead of ones derived from SMC on the current model. The full algorithm using the buffer is presented in Fig. 1 and Algorithm 4.

### 3.4. Practical Considerations

**Adaptive importance weight tempering.** When the proposal $\overrightarrow{p}_\theta$ is far from maintaining detailed balance with the flows $F_n^\phi$ between steps at which resampling is performed, the importance weights exhibit high variance: after self-normalisation, only a few samples receive non-negligible weights. Such degeneracy in turn increases the variance of gradient estimates, causing instability in the training. To address this issue, we temper the importance weights using an inverse temperature parameter $\lambda \in [0, 1]$, transforming $w \mapsto w^\lambda$ before normalisation (El-Laham et al., 2018).

While this could significantly reduce variance, it also introduces bias to our approximations. Thus, rather than using a fixed $\lambda$, we adaptively set $\lambda$ to maintain a minimum level of sample diversity, measured by the effective sample size (ESS). ESS represents the ratio of variances between plain Monte Carlo and self-normalised importance sampling estimators, commonly approximated as:

$$\widehat{\text{ESS}}(w^{1:K}) = \frac{\left( \sum_{k=1}^{K} w^k \right)^2}{\sum_{k=1}^{K} (w^k)^2}, \quad (14)$$

originally proposed by Kong (1992) and recently studied by Elvira et al. (2022) (see other alternative metrics in (Martino et al., 2017)). Note that the value $\widehat{\text{ESS}}$ ranges from 1 (complete degeneracy) to $K$ (equal weights).

Here, we propose a nonlinear transformation of the importance weights, an approach to reduce the variance of the weights (and the variance of the estimators as a consequence). Multiple variants of transformations have been explored in the literature, *e.g.*, clipping (Ionides, 2008; Koblents & Míguez, 2015) or adjusting the weights to follow a given distribution (Vehtari et al., 2024) (see a review in (Martino et al., 2018b)). Here, we perform an adaptive tempering scheme that smooths the weights by selecting the largest $\lambda$ that maintains $\widehat{\text{ESS}}$ above a specified threshold:

$$\lambda^* = \max \left\{ \lambda \in [0, 1] : \widehat{\text{ESS}} \left( (w^\lambda)^{1:K} \right) \geq \gamma K \right\}, \quad (15)$$

where $\gamma \in [0, 1]$ is a user-defined threshold. Since $\widehat{\text{ESS}} \left( (w^\lambda)^{1:K} \right)$ decreases monotonically from $K$ (when $\lambda = 0$) to $\widehat{\text{ESS}}(w^{1:K})$ (when $\lambda = 1$), we can efficiently find $\lambda^*$ using binary search with negligible computational overhead. See Algorithm 6.

As training progresses and $\overrightarrow{p}_\theta$ better approximates the target, we expect $\lambda^*$ to gradually increase toward 1, since the model learns to minimise the variance of importance weights (see §3.1).

**Adaptive resampling.** The SMC algorithm (§2.3) allows the choice of whether to perform resampling at each step $n$ in the propagation of particles. As is common in the literature (Del Moral et al., 2012), we use an adaptive scheme: resampling at step $n$ is performed only if the effective sample size, estimated from the current weights $w_n^k$ by (14), does not exceed some threshold $\kappa$. We also experiment with a scheme that uses only trajectory-level importance weighting (*i.e.*, AIS). See Algorithm 7 for the full algorithm.

**Diffusion-specific techniques.** In the case of diffusion samplers (considered in §4), we make use of improved training techniques proposed in past work: an expression of the intermediate flows as corrections to a temperature annealing scheme and a parametrisation of the proposals in terms of the gradient of the target energy. See §E for details.

## 4. Experiments

In this section, we primarily study diffusion samplers, as they provide well-established benchmarks for amortised sampling in continuous spaces. For our results on alanine dipeptide and discrete sample spaces, see §H and §I.

### 4.1. Diffusion Sampling Benchmarks

We validated our algorithm on a comprehensive suite of sampling benchmarks taken from Midgley et al. (2023); Sendera et al. (2024); Blessing et al. (2024); Chen et al. (2025). The evaluation is organised into two distinct settings:

- We first consider a *gradient-free* setting where the gradi-

*Table 1.* Benchmark results in the gradient-free setting. Here and in other tables, the best and second-best mean values are in **bold** and underlined, respectively, and we report mean and standard deviation over 5 runs.

| Target → | GMM40 ($d=2$) | | | GMM40 ($d=5$) | | | Funnel ($d=10$) | | | ManyWell ($d=32$) | | |
|---|---|---|---|---|---|---|---|---|---|---|---|---|
| Alg. ↓ Metric → | ELBO ↑ | EUBO ↓ | Sinkhorn ↓ | ELBO ↑ | EUBO ↓ | Sinkhorn ↓ | ELBO ↑ | EUBO ↓ | Sinkhorn ↓ | ELBO ↑ | EUBO ↓ | Sinkhorn ↓ |
| SMC-RWM | - | - | $41.17_{\pm 7.64}$ | - | - | $1787.5_{\pm 272.0}$ | - | - | $168.38_{\pm 1.45}$ | - | - | $29.14_{\pm 0.42}$ |
| DDS | $-2.36_{\pm 0.04}$ | $159.16_{\pm 25.17}$ | $595.54_{\pm 9.94}$ | $\underline{-5.16}_{\pm 0.02}$ | $3054.3_{\pm 84.9}$ | $3009.6_{\pm 0.8}$ | $\mathbf{-0.49}_{\pm 0.01}$ | $9.67_{\pm 1.54}$ | $136.85_{\pm 0.84}$ | $160.71_{\pm 0.02}$ | $259.08_{\pm 6.67}$ | $29.58_{\pm 0.02}$ |
| LV | $-2.36_{\pm 0.02}$ | $296.63_{\pm 46.23}$ | $616.78_{\pm 1.96}$ | $-5.19_{\pm 0.01}$ | $2936.0_{\pm 424.4}$ | $3059.8_{\pm 99.1}$ | $\underline{-0.50}_{\pm 0.01}$ | $8.66_{\pm 1.10}$ | $137.40_{\pm 0.88}$ | $160.70_{\pm 0.02}$ | $267.35_{\pm 4.83}$ | $29.58_{\pm 0.02}$ |
| TB | $\underline{-2.29}_{\pm 0.07}$ | $273.10_{\pm 45.62}$ | $607.31_{\pm 12.05}$ | $-5.19_{\pm 0.01}$ | $3156.7_{\pm 305.5}$ | $3110.2_{\pm 121.4}$ | $\underline{-0.50}_{\pm 0.01}$ | $8.33_{\pm 0.69}$ | $137.35_{\pm 0.84}$ | $160.69_{\pm 0.02}$ | $252.37_{\pm 6.00}$ | $29.57_{\pm 0.02}$ |
| + IW-Buf | $-2.37_{\pm 0.11}$ | $\mathbf{0.88}_{\pm 0.01}$ | $6.50_{\pm 0.11}$ | $\mathbf{-4.48}_{\pm 0.01}$ | $1183.3_{\pm 54.7}$ | $2813.9_{\pm 133.1}$ | $-0.70_{\pm 0.04}$ | $1.53_{\pm 0.37}$ | $\mathbf{115.28}_{\pm 3.45}$ | $\mathbf{162.84}_{\pm 0.04}$ | $\mathbf{166.30}_{\pm 0.02}$ | $\underline{22.97}_{\pm 0.02}$ |
| TB/SubTB + SMC | $\mathbf{-1.62}_{\pm 0.07}$ | $1.06_{\pm 0.01}$ | $39.99_{\pm 9.49}$ | $-10.56_{\pm 2.57}$ | $\underline{30.1}_{\pm 18.6}$ | $\underline{330.9}_{\pm 185.0}$ | $-0.56_{\pm 0.01}$ | $41.54_{\pm 15.75}$ | $138.79_{\pm 2.75}$ | $162.48_{\pm 0.05}$ | $166.83_{\pm 0.11}$ | $\mathbf{21.91}_{\pm 0.06}$ |
| + IW-Buf | $-2.44_{\pm 0.10}$ | $\underline{0.89}_{\pm 0.03}$ | $\mathbf{6.46}_{\pm 0.40}$ | $-11.46_{\pm 0.59}$ | $\mathbf{2.3}_{\pm 0.1}$ | $\mathbf{83.3}_{\pm 10.1}$ | $-0.69_{\pm 0.01}$ | $\underline{3.64}_{\pm 2.17}$ | $\underline{116.23}_{\pm 3.37}$ | $\underline{162.81}_{\pm 0.03}$ | $\mathbf{166.30}_{\pm 0.01}$ | $22.97_{\pm 0.04}$ |

*Table 2.* Benchmark results in the gradient-based setting. × indicates training instability (metric is above $10^5$). The results marked with * are directly from Chen et al. (2025).

| Target → | Funnel ($d=10$) | | Robot4 ($d=10$) | | GMM40 ($d=50$) | | MoS ($d=50$) | | ManyWell ($d=64$) | |
|---|---|---|---|---|---|---|---|---|---|---|
| Alg. ↓ Metric → | MMD ↓ | Sinkhorn ↓ | MMD ↓ | Sinkhorn ↓ | MMD ↓ | Sinkhorn ↓ | MMD ↓ | Sinkhorn ↓ | MMD ↓ | Sinkhorn ↓ |
| SMC-HMC | $0.197_{\pm 0.012}$ | $168.38_{\pm 1.45}$ | $0.645_{\pm 0.003}$ | $28.62_{\pm 0.03}$ | $0.486_{\pm 0.091}$ | $37836.18_{\pm 2390.97}$ | $0.326_{\pm 0.097}$ | $2320.79_{\pm 540.64}$ | $0.090_{\pm 0.005}$ | $75.79_{\pm 1.72}$ |
| SMC-HMC-ESS* | - | $117.48_{\pm 9.70}$ | - | $2.11_{\pm 0.31}$ | - | $24240.68_{\pm 50.52}$ | - | $\mathbf{1477.04}_{\pm 133.80}$ | - | - |
| PIS | $0.161_{\pm 0.002}$ | $155.16_{\pm 0.62}$ | $0.593_{\pm 0.026}$ | $3106.73_{\pm 684.05}$ | $0.110_{\pm 0.002}$ | $16425.73_{\pm 261.31}$ | $0.366_{\pm 0.025}$ | $2324.69_{\pm 81.66}$ | $0.244_{\pm 0.006}$ | $67.56_{\pm 0.99}$ |
| DDS | $0.116_{\pm 0.005}$ | $130.59_{\pm 2.10}$ | $1.381_{\pm 0.004}$ | × | $0.050_{\pm 0.001}$ | $6882.66_{\pm 125.25}$ | $0.245_{\pm 0.016}$ | $2170.75_{\pm 24.29}$ | $0.255_{\pm 0.000}$ | $66.41_{\pm 0.02}$ |
| LV | $0.109_{\pm 0.001}$ | $127.06_{\pm 0.77}$ | $0.422_{\pm 0.002}$ | $1.71_{\pm 0.01}$ | $\underline{0.036}_{\pm 0.000}$ | $3952.22_{\pm 97.14}$ | $0.350_{\pm 0.007}$ | $2175.86_{\pm 16.75}$ | $0.260_{\pm 0.000}$ | $66.65_{\pm 0.03}$ |
| CMCD-KL* | - | $124.89_{\pm 8.95}$ | - | $3.71_{\pm 1.00}$ | - | $22132.28_{\pm 595.18}$ | - | $1848.89_{\pm 532.56}$ | - | - |
| CMCD-LV* | - | $139.07_{\pm 9.35}$ | - | $27.00_{\pm 0.07}$ | - | $4258.57_{\pm 737.15}$ | - | $1945.71_{\pm 48.79}$ | - | - |
| SCLD w/o MCMC | $0.567_{\pm 0.079}$ | $226.49_{\pm 13.33}$ | $0.327_{\pm 0.007}$ | $1.28_{\pm 0.06}$ | $0.051_{\pm 0.025}$ | $6339.91_{\pm 4682.10}$ | $0.574_{\pm 0.016}$ | $3061.87_{\pm 313.92}$ | $0.263_{\pm 0.003}$ | $67.27_{\pm 0.52}$ |
| SCLD | $0.267_{\pm 0.004}$ | $169.38_{\pm 9.38}$ | $0.486_{\pm 0.033}$ | $11.24_{\pm 4.76}$ | $0.063_{\pm 0.003}$ | $7477.14_{\pm 489.84}$ | $\mathbf{0.130}_{\pm 0.002}$ | $\underline{1528.89}_{\pm 224.46}$ | $0.263_{\pm 0.001}$ | $67.27_{\pm 0.36}$ |
| TB | $0.109_{\pm 0.003}$ | $127.59_{\pm 1.10}$ | $0.424_{\pm 0.001}$ | $1.72_{\pm 0.01}$ | $0.036_{\pm 0.001}$ | $3903.95_{\pm 139.18}$ | $0.315_{\pm 0.023}$ | $2128.50_{\pm 64.94}$ | $0.243_{\pm 0.015}$ | $\underline{66.04}_{\pm 0.50}$ |
| + IW-Buf | $\underline{0.051}_{\pm 0.002}$ | $121.03_{\pm 2.75}$ | $\underline{0.318}_{\pm 0.003}$ | $1.27_{\pm 0.01}$ | $0.038_{\pm 0.001}$ | $4284.49_{\pm 170.43}$ | $0.442_{\pm 0.022}$ | $2505.42_{\pm 58.47}$ | $\underline{0.058}_{\pm 0.012}$ | $68.02_{\pm 0.71}$ |
| TB/SubTB + SMC | $0.073_{\pm 0.003}$ | $\mathbf{113.15}_{\pm 2.32}$ | $0.778_{\pm 0.339}$ | $64.48_{\pm 103.61}$ | $0.194_{\pm 0.315}$ | × | $0.428_{\pm 0.044}$ | $2416.52_{\pm 104.43}$ | $0.138_{\pm 0.013}$ | $\mathbf{63.77}_{\pm 0.18}$ |
| + IW-Buf | $\mathbf{0.050}_{\pm 0.008}$ | $\underline{115.54}_{\pm 1.97}$ | $\mathbf{0.103}_{\pm 0.110}$ | $\mathbf{0.39}_{\pm 0.44}$ | $\mathbf{0.035}_{\pm 0.001}$ | $\underline{3579.17}_{\pm 163.43}$ | $0.299_{\pm 0.043}$ | $2017.86_{\pm 131.56}$ | $\mathbf{0.043}_{\pm 0.003}$ | $68.57_{\pm 0.66}$ |

ent of the target log-density is unavailable. This prevents the use of gradient-based techniques such as Langevin parameterisation (LP; (Zhang & Chen, 2022)) and Langevin local search (Sendera et al., 2024), which makes the problem significantly more challenging (He et al., 2025). In this setting, we use several standard targets, including a 40-component Gaussian Mixture Model (GMM40) in $\mathbb{R}^2$ and $\mathbb{R}^5$, Funnel in $\mathbb{R}^{10}$ (Neal, 2003), and ManyWell in $\mathbb{R}^{32}$ (Noé et al., 2019).

- We then relax the black-box assumption (*gradient-based* setting) to benchmark against a wider range of baselines on more complex targets. We consider the Funnel in $\mathbb{R}^{10}$, GMM40 in $\mathbb{R}^{40}$, ManyWell in $\mathbb{R}^{64}$, a mixture of Student-t distributions (MoS) in $\mathbb{R}^{50}$ (Blessing et al., 2024), and Robot4 in $\mathbb{R}^{10}$, a robotics control problem used in Chen et al. (2025).

We train a diffusion sampler (§B.1) using our proposed off-policy schemes: sequential Monte Carlo with the learnt proposal and intermediate targets (SMC; §3.2) and importance-weighted experience replay (IW-Buf; §3.3). The training objective is the TB loss for the sampler $\overrightarrow{p}_\theta$, and SubTB for the flows $F_n^\phi$ when using SMC (see the further analysis of loss functions in §G.2). See §F for more details of the experimental setting.

**Baselines.** In the gradient-free setting, we consider SMC with random-walk Metropolis (SMC-RWM), and two diffusion sampling methods, DDS (Vargas et al., 2023) and log-variance (LV; Richter et al., 2020; 2024, similar to on-

policy TB). In the gradient-based setting, we include more advanced baselines that rely on the target gradients: SMC with Hamiltonian Monte Carlo (SMC-HMC) and its adaptive variant (SMC-HMC-ESS); CMCD (Vargas et al., 2024) with KL or log-variance losses (CMCD-KL, CMCD-LV); and SCLD (Chen et al., 2025). Note that DDS and PIS (Zhang & Chen, 2022) optimise reverse KL directly.

**Evaluation metrics.** We use the ELBO, EUBO, maximum mean discrepancy (MMD), and Sinkhorn distance (Cuturi, 2013) as evaluation metrics. Since comparing ELBO and EUBO across different generative processes is challenging (Blessing et al., 2024), we consider them only in the gradient-free setting, where all algorithms share the same diffusion process. The sample-based metrics are computed using the amortised sampler, *without* SMC. See §F.2 for more details on the metrics.

### 4.2. Main Results

We provide visualisations of generated samples in §G.1.

**Gradient-free setting.** The results are in Table 1. We observe that the on-policy algorithms (DDS, LV, and TB) all suffer severe mode collapse, which is expected due to the mode-seeking behaviour of the reverse-KL objectives.[4] The proposed SMC and IW-Buf significantly improve EUBO and Sinkhorn distance, indicating better mode coverage.

---

[4]On-policy TB and LV have the same expected gradient as the reverse-KL up to a multiplicative constant (Malkin et al., 2023).

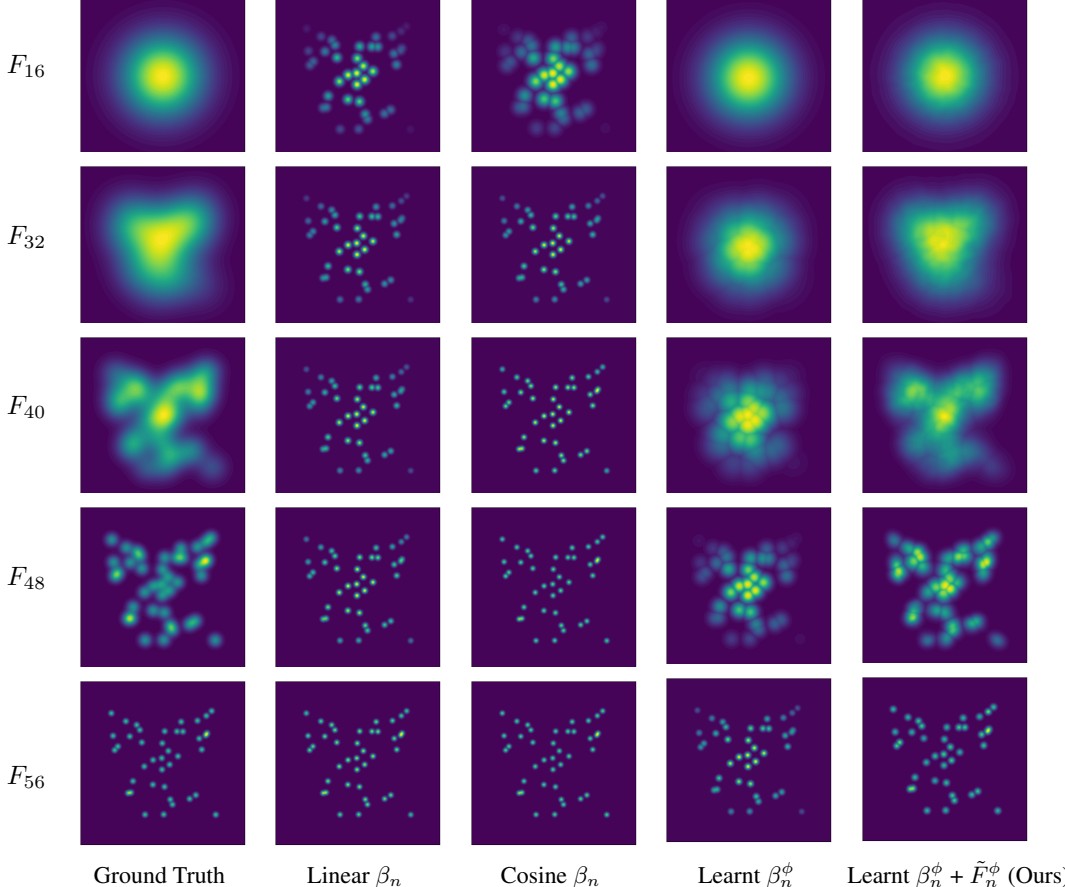

$F_{16}$

$F_{32}$

$F_{40}$

$F_{48}$

$F_{56}$

Ground Truth     Linear $\beta_n$     Cosine $\beta_n$     Learnt $\beta_n^\phi$     Learnt $\beta_n^\phi + \tilde{F}_n^\phi$ (Ours)

*Figure 2.* Visualisation of intermediate targets for GMM40 ($d = 2$) across different interpolation schemes. Note that $N = 64$, $F_0 = \overrightarrow{p}_0 = \mathcal{N}(0, \sigma^2 I)$, and $F_{64} = R$ is the unnormalised target density of GMM40 (defined in §F.1). The "Ground Truth" shows the exact intermediate target at each timestep, which can be derived analytically using $\pi \propto R$ and a fixed $\overleftarrow{p}$ and is also a GMM.

**Gradient-based setting.** As shown in Table 2, our proposed off-policy training schemes, SMC and IW-Buf, both improve performance. Integrating SMC as a behaviour policy is particularly effective for target distributions with sharp, separated modes, such as Robot4 or GMM40. Note that SCLD results here differ from those reported in Chen et al. (2025) due to a different evaluation protocol (see §F.3).

### 4.3. Analysis

**Effect of learnt flows $F_n^\phi$.** We investigate the effect of learning the intermediate marginals on our framework by comparing it against various interpolation schemes $\beta_n$ in (11). Based on geometric interpolation Equation (11), we consider linear, cosine, and learnt schedules of $\beta_n$. We test each both with and without a learnt correction that sets $F_n^\phi(\mathbf{x}_n) = \overrightarrow{p}_0(\mathbf{x}_n)^{1-\beta_n} R(\mathbf{x}_n)^{\beta_n} \tilde{F}_n^\phi(\mathbf{x}_n)$. Additionally, we include a baseline where the entire flow $F_n^\phi$ is learnt without interpolation. See §E for details.

In Fig. 2, we visualise the intermediate targets from different schemes for 2-dimensional GMM40 targets, where the ground-truth intermediate targets can be analytically com-

*Table 3.* Ablation study of learnable intermediate targets.

| Target $\rightarrow$ | ManyWell ($d = 32$) | | GMM40 ($d = 50$) | |
|---|---|---|---|---|
| $F_n \downarrow$ Metric $\rightarrow$ | ELBO $\uparrow$ | EUBO $\downarrow$ | MMD $\downarrow$ | Sinkhorn $\downarrow$ |
| Learnt $F_n^\phi$ (No interp.) | $161.52_{\pm 1.77}$ | $167.46_{\pm 1.92}$ | $0.040_{\pm 0.000}$ | $4622.4_{\pm 249.3}$ |
| Linear interp. | $162.57_{\pm 0.35}$ | $170.77_{\pm 4.10}$ | $\times$ | $\times$ |
| + Learnt $\tilde{F}_n^\phi$ | $162.32_{\pm 0.45}$ | $167.32_{\pm 1.90}$ | $0.039_{\pm 0.001}$ | $4675.9_{\pm 62.6}$ |
| Cosine interp. | $162.12_{\pm 0.75}$ | $169.96_{\pm 4.89}$ | $\times$ | $\times$ |
| + Learnt $\tilde{F}_n^\phi$ | $162.67_{\pm 0.15}$ | $168.12_{\pm 3.90}$ | $0.039_{\pm 0.000}$ | $4815.4_{\pm 311.4}$ |
| Learnt interp. | $162.48_{\pm 0.36}$ | $174.39_{\pm 8.96}$ | $0.041_{\pm 0.002}$ | $5091.1_{\pm 431.8}$ |
| + Learnt $\tilde{F}_n^\phi$ (Ours) | $\mathbf{162.81_{\pm 0.03}}$ | $\mathbf{166.30_{\pm 0.01}}$ | $\mathbf{0.035_{\pm 0.001}}$ | $\mathbf{3579.2_{\pm 163.4}}$ |

puted. The standard linear and cosine interpolations exhibit two major flaws: 1) they show drastic changes along the annealed path that deviate significantly from the ground truth; 2) they assign low density to modes located at the corners (because $\overrightarrow{p}_0$ is zero-mean Gaussian). When $\beta_n^\phi$ is learnt using SubTB, the annealed path becomes much smoother, though it still fails to fully resolve the latter. Our approach – which learns both the schedule and the correction – closely approximates the ground-truth intermediate targets across all timesteps.

Quantitative results on ManyWell ($d = 32$) and GMM40 ($d = 50$) targets are summarised in Table 3. We observe that training with SMC using fixed interpolation schedules

*Table 4.* Comparison to various buffer prioritisation schemes.

| Target → | GMM40 ($d = 5$) | | ManyWell ($d = 32$) | | GMM40 ($d = 50$) |
|---|---|---|---|---|---|
| Alg. ↓ Metric → | ELBO ↑ | EUBO ↓ | ELBO ↑ | EUBO ↓ | Sinkhorn ↓ |
| TB (on-policy) | $-5.19_{\pm 0.01}$ | $3156.7_{\pm 305.5}$ | $160.69_{\pm 0.02}$ | $252.37_{\pm 6.00}$ | $3904.0_{\pm 139.2}$ |
| + Buf | $-5.19_{\pm 0.02}$ | $3266.6_{\pm 1268.7}$ | $162.68_{\pm 0.13}$ | $170.06_{\pm 3.51}$ | $5088.1_{\pm 141.2}$ |
| + R-Buf | $-5.18_{\pm 0.02}$ | $2412.9_{\pm 216.4}$ | $161.23_{\pm 2.31}$ | $172.36_{\pm 3.51}$ | $5091.8_{\pm 135.8}$ |
| + L-Buf | $-5.18_{\pm 0.01}$ | $2488.3_{\pm 335.0}$ | $162.65_{\pm 0.09}$ | $173.26_{\pm 5.93}$ | $5018.5_{\pm 96.2}$ |
| + IW-Buf | $\mathbf{-4.48}_{\pm 0.01}$ | $1183.3_{\pm 54.7}$ | $\mathbf{162.84}_{\pm 0.04}$ | $166.30_{\pm 0.02}$ | $4284.5_{\pm 170.4}$ |
| TB/SubTB + SMC | $-10.56_{\pm 2.57}$ | $30.12_{\pm 18.63}$ | $162.48_{\pm 0.05}$ | $166.83_{\pm 0.11}$ | × |
| + Buf | $-4.78_{\pm 0.33}$ | $5062.6_{\pm 4758.2}$ | $162.80_{\pm 0.04}$ | $166.37_{\pm 0.03}$ | $3637.1_{\pm 170.6}$ |
| + R-Buf | $-4.52_{\pm 0.04}$ | $3037.9_{\pm 804.7}$ | $162.78_{\pm 0.01}$ | $166.38_{\pm 0.01}$ | $3752.4_{\pm 213.0}$ |
| + IW-Buf | $-11.46_{\pm 0.59}$ | $\mathbf{2.3}_{\pm 0.1}$ | $162.81_{\pm 0.03}$ | $\mathbf{166.30}_{\pm 0.01}$ | $\mathbf{3579.2}_{\pm 163.4}$ |

is unstable on complex target distributions, likely due to the high variance of the sequential importance weights (9), which leads to the degeneracy problem. In contrast, the flows are trained to minimise the variance of these weights, thereby mitigating this instability. Overall, training samplers with SMC using the learnt flows generally yields better samplers.

**Effect of importance-weighted buffer.** To assess the efficacy of our importance-weighted buffer (IW-Buf), we compare it against three baseline prioritisation schemes: a uniform buffer (Buf), a reward-prioritised buffer (R-Buf), and a loss-prioritised buffer (L-Buf). These schemes are tested both with and without SMC sampling, except for L-Buf, which cannot be combined with SMC since loss isn't defined for terminal samples alone. As shown in Table 4, the importance-weighting scheme outperforms the alternatives significantly, particularly in terms of mode coverage.

**Further analyses.** The additional analyses in §G include a study on loss function designs, the effect of adaptive tempering (§3.4), and a sensitivity analysis for key hyperparameters.

## 5. Conclusion

In this work, we have placed amortised sampling and sequential Monte Carlo in a common framework by building a dictionary between MaxEnt RL, hierarchical VI, and SMC. This has not been functionally done before, even though many connections among these have been noted and used. This dictionary allows a transfer of ideas and algorithms between these views on the sampling problem: SMC informs the design of an off-policy RL algorithm and the use of importance weights for experience replay; RL algorithms are used for the learning of SMC kernels and intermediate targets. Empirically, our proposed algorithms demonstrate substantial improvements over on-policy training.

Our work contributes to the growing body of research that connects learning-based sampling with Monte Carlo methods (Doucet et al., 2022; Chen et al., 2025; Albergo & Vanden-Eijnden, 2025). While sharing similarities with existing work, our approach is more general in that our framework does not assume any specific form in training loss, requiring only a general off-policy objective.

**Future work.** As discussed in §3.2, we used the unnormalised target density $R$ as the target for SMC used to obtain samples to train the proposal. There may exist better choices that adapt the target to favour regions that are more informative to the current sampler, *e.g.*, using the training loss along a trajectory as in Kim et al. (2025b). Future work should also consider methods other than SubTB for learning the intermediate targets, such as those used in twisted SMC (Lawson et al., 2018; 2022). In addition, the use of Monte Carlo corrector schemes (Del Moral et al., 2006) in the SMC proposal can be explored.

Finally, a large portion of this work focused on diffusion sampling in established low-dimensional benchmarks. Future work should apply the proposed methods to Bayesian posteriors over statistical model parameters (Blessing et al., 2024; Mittal et al., 2025, etc.), to discrete sampling problems in biological discovery (Jain et al., 2022), to the high-dimensional data-derived densities to which such samplers are beginning to be scaled (Venkatraman et al., 2025; Havens et al., 2025), and to the related problem of amortising Bayesian posterior inference under diffusion (Denker et al., 2024; Venkatraman et al., 2024; Domingo-Enrich et al., 2025) or other sequential generation priors (Hu et al., 2024; Zhao et al., 2024).

## Impact Statement

This paper presents work whose goal is to advance the field of Machine Learning. There are many potential societal consequences of our work, none of which we feel must be specifically highlighted here.

## Acknowledgements

The work of VE, ESW, and SC is supported by the Advanced Research and Invention Agency (ARIA). ESW acknowledges support from the CIFAR Learning in Machines and Brains programme. SM acknowledges funding from FRQNT Doctoral Fellowship (https://doi.org/10.69777/372208). SC and SM acknowledge computational resources provided by Mila.

The authors thank Oskar Kviman, Jarrid Rector-Brooks, and Siddarth Venkatraman for early discussions that inspired this project and Kirill Tamogashev for comments on a draft of the manuscript.

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

# A. Related Works

## A.1. Amortised Samplers and GFlowNets

Learning to sample from unnormalised distributions is a fundamental challenge in machine learning (Hinton, 2002). Early works from stochastic variational inference (Hoffman et al., 2013) rely on the REINFORCE estimator (Williams, 1992), which is often enhanced with advanced control variates (Titsias & Lázaro-Gredilla, 2014; Mnih & Gregor, 2014; Mnih & Rezende, 2016; Richter et al., 2020) to mitigate the high variance issue in REINFORCE.

Generative flow networks (GFlowNets; Bengio et al., 2021; 2023) were initially introduced as algorithms to learn policies that compositionally sample discrete objects proportionally to their rewards. GFlowNets bridge the gap between maximum entropy reinforcement learning (MaxEnt RL) algorithms (Haarnoja et al., 2017; 2018; Nachum et al., 2017) and amortised variational inference. Since MaxEnt RL methods learn a stochastic policy that samples actions proportionally to the expected soft returns *at each state*, they face a double-counting problem when multiple action sequences lead to the same object (Bengio et al., 2021), limiting their use as a general amortised inference method. GFlowNets effectively resolve this issue with a novel formulation of the Markov decision process (MDP) as a directed acyclic graph (DAG), enabling RL-like training of the amortised inference machine.

Subsequent research has established important connections between GFlowNets and (hierarchical) variational inference (Malkin et al., 2023; Zimmermann et al., 2023), as well as with MaxEnt RL algorithms (Tiapkin et al., 2024; Mohammadpour et al., 2024; Deleu et al., 2024). Additionally, GFlowNets have been generalised to continuous spaces (Lahlou et al., 2023), and a recent study has studied their connections to various continuous-time objectives (Berner et al., 2026). Collectively, these findings have positioned GFlowNets as general off-policy RL algorithms suitable for training amortised samplers.

The importance of off-policy training for amortised samplers has been particularly emphasised in GFlowNet literature. Replay buffers have demonstrated improved sample efficiency and target approximation in both discrete and continuous domains (Deleu et al., 2022; Tiapkin et al., 2024; Vemgal et al., 2023; Sendera et al., 2024, among others). Several Monte Carlo methods have been integrated into GFlowNet training, including Monte Carlo tree search for forward exploration (Morozov et al., 2024) and Monte Carlo exploration in the target space (Zhang et al., 2022; Kim et al., 2024b; Sendera et al., 2024; Kim et al., 2025b) as a guided exploration. These approaches provide valuable off-policy training examples that improve training efficiency and/or mode coverage.

In the next section, we provide a detailed review of works on diffusion samplers for continuous space, which have recently gained more attention than their discrete counterparts.

## A.2. Diffusion Samplers

Inspired by the successes of diffusion models in approximating empirical distributions by denoising processes (Sohl-Dickstein et al., 2015; Ho et al., 2020; Song et al., 2021b), diffusion samplers, beginning with Vargas et al. (2022); Zhang & Chen (2022); Vargas et al. (2023), are generative models of the same form that are trained to sample a distribution that can be queried for its energy (and possibly its gradient), but not sampled from directly. The problem of training a diffusion sampler is known to have an interpretation as stochastic optimal control (Zhang & Chen, 2022; Nüsken & Richter, 2021; Berner et al., 2024).

Diffusion samplers can be trained by off-policy methods, ones that minimise some discrepancy between a generative process and its reverse without differentiating through the sampling process. Such methods were independently introduced from the path space measure perspective (Richter et al., 2020; 2024) and that of GFlowNets in the continuous space in works including Lahlou et al. (2023) (further developed in Zhang et al. (2024); Sendera et al. (2024); Kim et al. (2025b)). A unifying perspective on all objectives and continuous-time limit analysis is provided in Berner et al. (2026). Various off-policy behaviour policies for diffusion samplers have been proposed: these policies attempt to modify the distribution of states seen during training so as to lead to better coverage of the modes of the target distribution by the trained sampler. Behaviour policies using auxiliary Monte Carlo exploration (Sendera et al., 2024) and an additional trained policy guided by the loss of the sampler (Kim et al., 2025b) show promise in this regard. Our simple and principled approach of using Monte Carlo methods for training trajectory selection is inspired by these works and shows further improvements in the training of diffusion samplers.

Several works have explored the connection between diffusion models or samplers and sequential Monte Carlo or importance sampling, albeit in different ways than this paper. For diffusion models, Monte Carlo methods, including SMC, have been

proposed in the setting of sampling a posterior distribution under a diffusion prior (Doucet et al., 2022; Cardoso et al., 2024; Song et al., 2023; Dou & Song, 2024; Kim et al., 2025d; Singhal et al., 2025; Skreta et al., 2025; He et al., 2026). These sampling algorithms run at inference time, rather than guiding the training of an amortised sampler that can provide unbiased samples in finite time at convergence. In Máté & Fleuret (2023), the training objective for a denoiser maintains the sampler in balance with (learnt) intermediate target distributions, amounting to an amortised adaptive proposal for twisted SMC in the continuous-time limit; related ideas are explored in Vargas et al. (2024); Chen et al. (2025); Albergo & Vanden-Eijnden (2025). However, these methods are typically coupled to a specialised training objective, while we are interested in behaviour policy selection for general off-policy training losses. In discrete spaces, SMC has been used for sampling intractable posterior distributions under language model priors (Lew et al., 2023; Loula et al., 2025).

### A.3. Optimised Sequential Monte Carlo

A prominent line of research focuses on choosing optimal kernels and intermediate targets to optimise SMC, *i.e.*, to minimise the variance of incremental importance weights (9) at each step (see, *e.g.*, Del Moral et al. (2006); Naesseth et al. (2019)). Guarniero et al. (2017) introduced the optimal form of twist for transition kernels for particle filters and used a dynamic programming approach to approximate it. Heng et al. (2020) later extended this to sampling from a static distribution. These approaches require a set of parametric assumptions in transition kernels or intermediate marginals to approximate the (generally intractable) integrals required for optimality. Syed et al. (2024) optimised the annealing schedule for intermediate distributions, while keeping the kernels invariant to intermediate targets. More recently, Zhao et al. (2024) proposed a contrastive learning approach to optimise twist functions for intermediate marginals, but their method is restricted to specific Markov chains in autoregressive models. Our framework treats a more general case, as it builds a connection between MaxEnt RL and SMC, accommodates general hierarchical latent variable models, and does not require any parametric assumptions on either the kernels or the intermediate marginals.

## B. Generative Processes for Diffusion and Sequence Generation

### B.1. Diffusion Models

Diffusion models (Ho et al., 2020; Song et al., 2021b) assume a generative process governed by a stochastic differential equation (SDE):

$$d\mathbf{y}_t = u_\theta(\mathbf{y}_t, t)dt + \sigma(t)d\mathbf{w}_t, \quad t \in [0, 1], \tag{16}$$

where $\mathbf{y}_0 \sim \overrightarrow{p}_0$ and $\mathbf{w}_t$ is standard Brownian motion. The goal is to learn the drift $u_\theta$ such that the marginal distribution of $\mathbf{y}_1 \in \mathcal{X}$ closely approximates the target $\pi$.

In practice, we often discretise the SDE via *Euler-Maruyama* scheme (Maruyama, 1955) with predefined time points $0 = t_0 < t_1 < \cdots < t_N = 1$. This yields a discrete-time Markov process with transition kernel:

$$\overrightarrow{p}_\theta(\mathbf{y}_{t_{n+1}}|\mathbf{y}_{t_n}) = \mathcal{N}(\mathbf{y}_{t_{n+1}}; \mathbf{y}_{t_n} + u_\theta(\mathbf{y}_{t_n}, t_n)\Delta t_n, \sigma^2(t_n)\Delta t_n \mathbf{I}_d), \tag{17}$$

where $\Delta t_n = t_{n+1} - t_n$. By setting the latent variables as $\mathbf{x}_n = \mathbf{y}_{t_n}$, we can see that the diffusion models belong to the family of hierarchical latent variable models.

Note that we used a different discretisation scheme for our diffusion sampling experiments (§4); see §F.4 for details.

### B.2. Prepend/Append Models

We present the prepend/append sequence generation process introduced in (Shen et al., 2023) by comparing it to autoregressive models. In the sequence generation setting, $x \in \mathcal{X}$ is a complete sequence with length $N$ and $\mathbf{x}_n$ is a string that consists of $n$ characters from a predefined vocabulary $\mathcal{V}$ ($\mathbf{x}_0$ is an empty string), *i.e.*, $\mathbf{x}_n = [v_1, \ldots, v_n]$ where $v_i \in \mathcal{V}$ for all $i = 1, \ldots, n$.

Autoregressive models permit only the append operation, so the conditional distribution is defined as $\overrightarrow{p}_\theta(\mathbf{x}_n|\mathbf{x}_{n-1}) = \overrightarrow{p}_\theta(\mathbf{x}_{n-1} \oplus [v]|\mathbf{x}_{n-1}) = \overrightarrow{p}_\theta^{\text{app}}(v|\mathbf{x}_{n-1})$, where $\oplus$ is the append operator. In autoregressive generation, there exists only one trajectory that leads to any $\mathbf{x}$.

In contrast, the prepend/append models define the conditional distribution as:

$$
\begin{aligned}
\overrightarrow{p}_\theta(\mathbf{x}_n|\mathbf{x}_{n-1}) \propto\ & \mathbb{I}(\mathbf{x}_n = [v] \oplus \mathbf{x}_{n-1})\overrightarrow{p}_\theta^{\text{prep}}(v|\mathbf{x}_{n-1}) \\
& + \mathbb{I}(\mathbf{x}_n = \mathbf{x}_{n-1} \oplus [v])\overrightarrow{p}_\theta^{\text{app}}(v|\mathbf{x}_{n-1}),
\end{aligned}
\tag{18}
$$

where $\mathbb{I}$ is the indicator function, and $\overrightarrow{p}_\theta^{\text{prep}}$ and $\overrightarrow{p}_\theta^{\text{app}}$ are probability distributions over tokens to prepend and append, respectively. This formulation allows the model to build sequences from both ends, providing greater flexibility than purely autoregressive approaches. Unlike the autoregressive case, there exist at most $2^{N-1}$ trajectories that end in an $\mathbf{x}$ with length $N$.

## C. Training Loss

Upon Equation (7), Madan et al. (2023) proposed a scheme to give weights to subtrajectories in a whole trajectory $\mathbf{x}_{0:N} = (\mathbf{x}_0, \mathbf{x}_1 \ldots, \mathbf{x}_N)$:

$$
\mathcal{L}_{\text{SubTB-}\lambda}^{\theta,\phi}(\mathbf{x}_{0:N}) = \frac{\sum_{0 \leq m < n \leq N} \lambda^{n-m} \mathcal{L}_{\text{SubTB}}^{\theta,\phi}(\mathbf{x}_{m:n})}{\sum_{0 \leq m < n \leq N} \lambda^{n-m}},
\tag{19}
$$

where $\lambda > 0$ is a hyperparameter.

In this work, we propose an alternative approach to combining subtrajectory objectives, which we refer to as chunk-based SubTB. First, let $L$ be the length of each chunked subtrajectory. For simplicity, assume $N$ is a multiple of $L$, i.e., $N/L \in \mathbb{N}$. We divide $\mathbf{x}_{0:N}$ into non-overlapping $N/L$ chunks $\{\mathbf{x}_{0:L}, \mathbf{x}_{L:2L}, \ldots, \mathbf{x}_{N-L:N}\}$. The chunk-based SubTB loss is then defined as:

$$
\mathcal{L}_{\text{SubTB-Chunk}(L)}^{\theta,\phi}(\mathbf{x}_{0:N}) = \sum_{i=0}^{N/L-1} \left( \underbrace{\mathcal{L}_{\text{SubTB}}^{\theta,\phi}(\mathbf{x}_{iL:(i+1)L})}_{(1)} + \underbrace{\frac{1}{N/L-i}\mathcal{L}_{\text{SubTB}}^{\theta,\phi}(\mathbf{x}_{iL:N})}_{(2)} \right).
\tag{20}
$$

Note that the sum of (1) is analogous to the detailed balance objective (Bengio et al., 2023) but applied to chunked subtrajectories. The term (2) significantly stabilises optimisation by mitigating bootstrapping error in (1), since it always includes the terminal reward $F^\phi(\mathbf{x}_N) = R(\mathbf{x}_N)$ which is not learnt.

Our chunk-based objective, $\mathcal{L}_{\text{SubTB-Chunk}(L)}$, achieves better empirical results compared to $\mathcal{L}_{\text{SubTB-}\lambda}$ (see §G.2). Moreover, while $\mathcal{L}_{\text{SubTB-}\lambda}$ requires $N$ evaluations of $F^\phi$, the chunk-based SubTB only needs $N/L$ evaluations, which can be particularly more efficient when these evaluations are expensive.

## D. Algorithms

---

**Algorithm 1** Importance-weighted Training

---

**Require:** The unnormalised target $R$, reverse kernel $\overleftarrow{p}$, initial distribution $\overrightarrow{p}_0$, forward kernel model $\overrightarrow{p}_\theta$, number of training epochs $N_{\text{epoch}}$, off-policy ratio $I$, batch size $K$, number of discretisation steps $N$, ess threshold for adaptive tempering $\gamma$.

1: **for** $i = 1, \ldots, N_{\text{epoch}}$ **do**
2:     Sample $K$ trajectories and compute their weight using $\overrightarrow{p}_0$ and $\overrightarrow{p}_\theta \Rightarrow \{\mathbf{x}_{0:N}^k, w_N^k\}_{k=1}^K$.
3:     **if** $i \bmod I = 0$ **then**
4:         $W_N^k = 1/K$ for all $k$.
5:     **else**
6:         Obtain adaptive inverse temperature, $\lambda \leftarrow \text{ADAPTIVEIWTEMPERING}(\{w_N^k\}_{k=1}^K, \gamma)$.         $\triangleright$ Algorithm 5
7:         Calculate the self-normalised importance weight $W_N^k = \frac{(w_N^k)^\lambda}{\sum_l (w_N^l)^\lambda}$ for each $\mathbf{x}_{0:N}^k$.
8:     **end if**
9:     $\theta \leftarrow \text{Optimiser}\left(\theta, \nabla_\theta\left(\sum_{k=1}^K W_N^k \mathcal{L}_{\text{TB}}^\theta(\mathbf{x}_{0:N}^k)\right)\right)$.         $\triangleright$ Equation (8)
10: **end for**

---

**Algorithm 2** Training with sequential Monte Carlo

---

**Require:** The unnormalised target $R$, reverse kernel $\overleftarrow{p}$, initial distribution $\overrightarrow{p}_0$, forward kernel model $\overrightarrow{p}_\theta$, flow function model $F^\phi$, number of training epochs $N_{\text{epoch}}$, off-policy ratio $I$, batch size $K$, number of discretisation steps $N$, subtrajectory length $L$, ESS threshold for adaptive resampling $\kappa$, ESS threshold for adaptive tempering $\gamma$.
**Ensure:** $N/L \in \mathbb{N}$.

1: **for** $i = 1, \ldots, N_{\text{epoch}}$ **do**
2:     **if** $i \bmod I = 0$ **then**         $\triangleright$ On-policy sampling
3:         Sample $K$ forward trajectories using $\overrightarrow{p}_0$ and $\overrightarrow{p}_\theta \Rightarrow \{\mathbf{x}_{0:N}^k\}_{k=1}^K$.
4:     **else**         $\triangleright$ Off-policy sampling
5:         Run SMC, $\{\mathbf{x}_N^k, \bar{w}_N^k\}_{k=1}^K \leftarrow \text{SMCSAMPLING}(R, \overleftarrow{p}, \overrightarrow{p}_0, \overrightarrow{p}_\theta, F^\phi, K, N, L, \kappa, \gamma)$.     $\triangleright$ Algorithm 7
6:         For each $k$, sample a backward trajectory from $\mathbf{x}_N^k$ using $\overleftarrow{p} \Rightarrow \{\mathbf{x}_{0:N}^k\}_{k=1}^K$.
7:     **end if**
8:     $\phi \leftarrow \text{Optimiser}\left(\phi, \nabla_\phi\left(\sum_{k=1}^K \mathcal{L}_{\text{SubTB-Chunk}(L)}^{\theta,\phi}(\mathbf{x}_{0:N}^k)\right)\right)$.         $\triangleright$ Equation (20)
9:     $\theta \leftarrow \text{Optimiser}\left(\theta, \nabla_\theta\left(\sum_{k=1}^K \mathcal{L}_{\text{TB}}^\theta(\mathbf{x}_{0:N}^k)\right)\right)$.         $\triangleright$ Equation (8)
10: **end for**

---

---

**Algorithm 3** Training with the Importance-weighted Experience Replay

---

**Require:** The unnormalised target $R$, reverse kernel $\overleftarrow{p}$, initial distribution $\overrightarrow{p}_0$, forward kernel model $\overrightarrow{p}_\theta$, number of training epochs $N_{\text{epoch}}$, off-policy ratio $I$, batch size $K$, number of discretisation steps $N$, ESS threshold for adaptive tempering $\gamma$.
1: Initialise $\mathcal{B} = \emptyset$.
2: **for** $i = 1, \ldots, N_{\text{epoch}}$ **do**
3:     **if** $i \bmod I = 0$ **then**                                                  ▷ On-policy sampling
4:         Sample $K$ forward trajectories and compute their weight using $\overrightarrow{p}_0$ and $\overrightarrow{p}_\theta \Rightarrow \{\mathbf{x}_{0:N}^k, w_N^k\}_{k=1}^K$.
5:         Add the terminal states and weights in the buffer, $\mathcal{B} \leftarrow \mathcal{B} \cup \{\mathbf{x}_N^k, w_N^k\}_{k=1}^K$.
6:     **else**                                                                ▷ Off-policy sampling
7:         Obtain adaptive inverse temperature, $\lambda \leftarrow \text{ADAPTIVEIWTEMPERING}(\{w_N^l\}_{l=1}^{|\mathcal{B}|}, \gamma)$.         ▷ Algorithm 5
8:         Draw $K$ terminal states from $\mathcal{B}$ by sampling proportionally to $(w_N^l)^\lambda \Rightarrow \{\mathbf{x}_N^k\}_{k=1}^K$.
9:         For each $k$, sample a backward trajectory from $\mathbf{x}_N^k$ using $\overleftarrow{p} \Rightarrow \{\mathbf{x}_{0:N}^k\}_{k=1}^K$.
10:     **end if**
11:     $\theta \leftarrow \text{Optimiser}\left(\theta, \nabla_\theta\left(\sum_{k=1}^K W_N^k \mathcal{L}_{\text{TB}}^\theta(\mathbf{x}_{0:N}^k)\right)\right)$.                             ▷ Equation (8)
12: **end for**

---

**Algorithm 4** Training with both sequential Monte Carlo and Importance-weighted Experience Replay

---

**Require:** The unnormalised target $R$, reverse kernel $\overleftarrow{p}$, initial distribution $\overrightarrow{p}_0$, forward kernel model $\overrightarrow{p}_\theta$, flow function models $F^\phi$, number of training epochs $N_{\text{epoch}}$, off-policy ratio $I$, batch size $K$, number of discretisation steps $N$, subtrajectory length $L$, ESS threshold for adaptive resampling $\kappa$, ESS threshold for adaptive tempering $\gamma$.
**Ensure:** $N/L \in \mathbb{N}$.
1: Initialise $\mathcal{B} = \emptyset$.
2: **for** $i = 1, \ldots, N_{\text{epoch}}$ **do**
3:     **if** $i \bmod I = 0$ **then**                                                  ▷ On-policy sampling
4:         Sample $K$ forward trajectories and compute their weight using $\overrightarrow{p}_0$ and $\overrightarrow{p}_\theta \Rightarrow \{\mathbf{x}_{0:N}^k, w_N^k\}_{k=1}^K$.
5:         Let $\bar{w}_N^k = w_N^k$.
6:         Add the terminal states and weights in the buffer, $\mathcal{B} \leftarrow \mathcal{B} \cup \{\mathbf{x}_N^k, \bar{w}_N^k\}_{k=1}^K$.
7:     **else**                                                                 ▷ Off-policy sampling
8:         Run SMC, $\{\mathbf{x}_N^j, \bar{w}_N^j\}_{j=1}^K \leftarrow \text{SMCSAMPLING}(R, \overleftarrow{p}, \overrightarrow{p}_0, \overrightarrow{p}_\theta, F^\phi, K, N, L, \kappa, \gamma)$.     ▷ Algorithm 7
9:         Add the terminal states and weights in the buffer, $\mathcal{B} \leftarrow \mathcal{B} \cup \{\mathbf{x}_N^j, \bar{w}_N^j\}_{j=1}^K$.
10:         Obtain adaptive inverse temperature, $\lambda \leftarrow \text{ADAPTIVEIWTEMPERING}(\{\bar{w}_N^l\}_{l=1}^{|\mathcal{B}|}, \gamma)$.     ▷ Algorithm 5
11:         Draw $K$ terminal states from $\mathcal{B}$ by sampling proportionally to $(\bar{w}_N^l)^\lambda \Rightarrow \{\mathbf{x}_N^k\}_{k=1}^K$.
12:         For each $k$, sample a backward trajectory from $\mathbf{x}_N^k$ using $\overleftarrow{p} \Rightarrow \{\mathbf{x}_{0:N}^k\}_{k=1}^K$.
13:     **end if**
14:     $\phi \leftarrow \text{Optimiser}\left(\phi, \nabla_\phi\left(\sum_{k=1}^K \mathcal{L}_{\text{SubTB-Chunk}(L)}^{\theta,\phi}(\mathbf{x}_{0:N}^k)\right)\right)$.             ▷ Equation (20)
15:     $\theta \leftarrow \text{Optimiser}\left(\theta, \nabla_\theta\left(\sum_{k=1}^K \mathcal{L}_{\text{TB}}^\theta(\mathbf{x}_{0:N}^k)\right)\right)$.                        ▷ Equation (8)
16: **end for**

---

**Algorithm 5** Adaptive Importance Weight Tempering

---

1: **function** ADAPTIVEIWTEMPERING($\{w^k\}_{k=1}^K, \gamma$)
2:     Initialise $\lambda^* = 1.0$.
3:     **if** $\widehat{\text{ESS}}(\{w^k\}_{k=1}^K) < \gamma K$ **then**
4:         Binary Search for $\lambda^*$ that satisfies Equation (15).
5:     **end if**
6:     **return** $\lambda^*$
7: **end function**

---

---

**Algorithm 6** Resampling with Adaptive Tempering

---

1: **function** RESAMPLING($\{\mathbf{x}^k, w^k\}_{k=1}^K, \gamma$)
2:      Obtain adaptive inverse temperature, $\lambda \leftarrow$ ADAPTIVEIWTEMPERING($\{w^k\}_{k=1}^K, \gamma$).      ▷ Algorithm 5
3:      $\{a(k)\}_{k=1}^K \leftarrow$ Sample indices proportionally to $\frac{(w^k)^\lambda}{\sum_l (w^l)^\lambda}$.
4:      For each $k$, $\check{\mathbf{x}}^k = \mathbf{x}^{a(k)}$ and $\check{w}^k = \frac{(w^{a(k)})^{1-\lambda}}{\sum_l (w^{a(l)})^{1-\lambda}}$.
5:      **return** $\{\check{\mathbf{x}}^k, \check{w}^k\}_{k=1}^K$
6: **end function**

---

**Algorithm 7** Sequential Monte Carlo Sampling

---

1: **function** SMCSAMPLING($R, \overleftarrow{p}, \overrightarrow{p}_0, \overrightarrow{p}_\theta, F^\phi, K, N, L, \kappa, \gamma$)
2:      Initialise particles and weights $\{\mathbf{x}_0^k, w_0^k\}_{k=1}^K$, where $\mathbf{x}_0^k \sim \overrightarrow{p}_0$ and $w_0^k = 1/K$ for all k.
3:      Initialise $\hat{Z}_0 = 0$.
4:      **for** $j = 1, \ldots, N/L$ **do**
5:          Sample subtrajectories from $\{\mathbf{x}_{(j-1)L}^k\}_{k=1}^K$ using $\overrightarrow{p}_\theta \Rightarrow \{\mathbf{x}_{(j-1)L:jL}^k\}_{k=1}^K$
6:          Update weights for each subtrajectories using $\overleftarrow{p}, \overrightarrow{p}_\theta$ and $F^\phi \Rightarrow \{\mathbf{x}_{jL}^k, w_{jL}^k\}_{k=1}^K$.      ▷ Equation (9)
7:          $\hat{Z}_{jL} \leftarrow \hat{Z}_{(j-1)L} \cdot \sum_{k=1}^K w_{jL}^k$.
8:          **if** $\widehat{\text{ESS}}\left(\{w_{jL}^k\}_{k=1}^K\right) < \kappa K$ and $j < N/L$ **then**
9:              $\{\mathbf{x}_{jL}^k, w_{jL}^k\}_{k=1}^K \leftarrow$ RESAMPLING($\{\mathbf{x}_{jL:jL}^k, w_{jL}^k\}_{k=1}^K, \gamma$).      ▷ Algorithm 6
10:         **else**
11:              $\{\mathbf{x}_{jL}^k, w_{jL}^k\}_{k=1}^K \leftarrow \{\mathbf{x}_{jL}^k, \frac{w_{jL}^k}{\sum_l w_{jL}^l}\}_{k=1}^K$.
12:         **end if**
13:      **end for**
14:      $\bar{w}_N^k = K \cdot \hat{Z}_N \cdot w_N^k$.
15:      **return** $\{\mathbf{x}_N^k, \bar{w}_N^k\}_{k=1}^K$
16: **end function**

---

## E. Training Techniques for Diffusion Samplers

We use two methods from the literature on diffusion samplers to improve learning in this setting: Langevin parametrisation and a partial energy (reward shaping) scheme.

**Langevin parametrisation.** In a diffusion sampler, the forward policy is expressed in terms of the drift $u_\theta$ of an underlying neural SDE (see (17)). Whereas the default approach would express $u_\theta(\mathbf{x}_t, t)$ as a neural network taking $\mathbf{x}_t$ and $t$ as input, the Langevin parametrisation, first used by Zhang & Chen (2022), writes this drift in terms of the score function of the target density:

$$u_\theta(\mathbf{x}_t, t) = \text{NN}_1^\theta(\mathbf{x}_t, t) + \text{NN}_2^\theta(t; \theta) \nabla \log R(\mathbf{x}_t), \tag{21}$$

where $\text{NN}_1^\theta$ and $\text{NN}_2^\theta$ are neural networks with vector and scalar output, respectively. Such a method expresses the drift as an additive correction to Langevin dynamics on the target density and is expected to guide the sampler to regions of high density.

**Partial energy.** For convenience, we restate the geometric interpolation from (11) here:

$$F_n(\mathbf{x}) = \overrightarrow{p}_0(\mathbf{x})^{1-\beta_n} R(\mathbf{x})^{\beta_n}.$$

Beyond being the standard choice for defining intermediate targets in SMC, this interpolation has been used as an inductive bias for learnt flows (unnormalised intermediate targets) in amortised samplers. Specifically, Máté & Fleuret (2023); Zhang et al. (2024) considered a learnable correction to a simple geometric interpolation, and Vargas et al. (2024); Chen et al. (2025) made the annealing schedule learnable $\beta_n^\phi$. Building on these ideas, we combine both approaches to parameterise $F_n^\phi$ as follows:

$$F_n^\phi(\mathbf{x}_n) = \overrightarrow{p}_0(\mathbf{x}_n)^{1-\beta_n^\phi} R(\mathbf{x}_n)^{\beta_n^\phi} \tilde{F}_n^\phi(\mathbf{x}_n), \tag{22}$$

where $\beta_n^\phi$ is a learnt annealing schedule and $\tilde{F}_n^\phi$ is the learnable correction. To define $\beta_n^\phi$, we introduce $N$ learnable scalars, $\{\phi_n\}_{n=1}^N$, and compute the schedule via $\beta_n^\phi = \sum_{i=1}^n \frac{\text{Softplus}(\phi_i)}{\sum_{j=1}^N \text{Softplus}(\phi_j)}$, where $\text{Softplus}(x) = \log(1 + \exp(x))$.

## F. Detailed Experimental Settings

In this section, we formally define the synthetic target distributions and provide details about the model architecture and the hyperparameters used for experiments in §4.

### F.1. Target Distributions

- **GMM40** ($d \in [2, 5, 50]$) (Midgley et al., 2023) is a $d$-dimensional mixture of Gaussians with 40 components. Each component $i$ has mean $\boldsymbol{\nu}_i \in \mathbb{R}^d$, where each dimension of it is randomly sampled from $\mathcal{U}[-40, 40]$, and shares an identity covariance matrix $I$. The unnormalised density is then defined as:

$$R_{\text{GMM40}}(\mathbf{x}) = \frac{1}{40} \sum_{i=1}^{40} \mathcal{N}(\mathbf{x}; \boldsymbol{\nu}_i, I).$$

- **Funnel** ($d = 10$) (Neal, 2003) is a funnel-shaped distribution, whose unnormalised density is defined as:

$$R_{\text{Funnel}}(\mathbf{x}) = \mathcal{N}(x_1; 0, 3^2) \prod_{i=2}^{10} \mathcal{N}(x_i; 0, \exp(x_1)),$$

where $\mathbf{x} = (x_i)_{i=1}^{10}$.

- **ManyWell** ($d \in [32, 64]$) (Nüsken & Richter, 2021; Midgley et al., 2023) is constructed as the product of $d/2$ independent copies of a 2-dimensional Double Well distribution (Noé et al., 2019). The Double Well energy is:

$$\mathcal{E}_{\text{DW}}(x_1, x_2) = x_1^4 - 6x_1^2 - \frac{1}{2}x_1 + \frac{1}{2}x_2^2.$$

The unnormalised density of ManyWell is then defined as:

$$R_{\text{ManyWell}}(\mathbf{x}) = \exp\left(-\sum_{i=1}^{d/2} \mathcal{E}_{\text{DW}}(x_{2i-1}, x_{2i})\right).$$

Ground truth samples are obtained via rejection sampling.

- **Robot4** ($d = 10$) (Arenz et al., 2020; Chen et al., 2025) defines the distribution over joint configurations of a planar robot with 10 degrees of freedom. The target unnormalised density is defined as:

$$R_{\text{Robot4}}(\mathbf{x}) = R_{\text{conf}}(\mathbf{x})R_{\text{cart}}(\mathbf{x}),$$

where $R_{\text{conf}}$ encourages smooth configurations by placing a Gaussian prior and $R_{\text{cart}}$ attracts the robot's end-effector to one of four possible goal locations:

$$R_{\text{conf}}(\mathbf{x}) = \mathcal{N}(x_1; 0, 1^2) \prod_{i=2}^{10} \mathcal{N}(x_i; 0, (0.2)^2)$$

$$R_{\text{cart}}(\mathbf{x}) = \max_{j \in \{1,2,3,4\}} \mathcal{N}\left(\begin{pmatrix} a(\mathbf{x}) \\ b(\mathbf{x}) \end{pmatrix}; \begin{pmatrix} a_g^j \\ b_g^j \end{pmatrix} \begin{pmatrix} 10^{-4} & 0 \\ 0 & 10^{-4} \end{pmatrix}\right),$$

where $a(\mathbf{x})$ and $b(\mathbf{x})$ are end-effector position of robot calculated as $a(\mathbf{x}) = \sum_{i=1}^{10} \cos(x_i)$ and $b(\mathbf{x}) = \sum_{i=1}^{10} \sin(x_i)$, and the goal positions $\begin{pmatrix} a_g^j \\ b_g^j \end{pmatrix}$ are $\begin{pmatrix} 7 \\ 0 \end{pmatrix}, \begin{pmatrix} -7 \\ 0 \end{pmatrix}, \begin{pmatrix} 0 \\ 7 \end{pmatrix}$, and $\begin{pmatrix} 0 \\ -7 \end{pmatrix}$. Ground truth samples are taken from (Arenz et al., 2020).

- **MoS** ($d = 50$) (Blessing et al., 2024) is a $d$-dimensional mixture of Student's t-distributions with 10 components. Let $t_2$ be the density of Student's t-distribution with degree 2 and $\boldsymbol{\nu}_i$ be the shift of component $i$. Each dimension of $\boldsymbol{\nu}_i$ is randomly sampled from $\mathcal{U}[-10, 10]$. The unnormalised density of MoS is defined as:

$$R_{\text{MoS}}(\mathbf{x}) = \frac{1}{10} \sum_{i=1}^{10} t_2(\mathbf{x} - \boldsymbol{\nu}_i).$$

## F.2. Evaluation Metrics

In this section, we introduce the evaluation metrics used to benchmark the diffusion samplers. We employ three distribution-based metrics: the evidence lower bound (ELBO), importance-weighted ELBO (IW-ELBO), and evidence upper bound (EUBO; Blessing et al., 2024). Also, following Chen et al. (2025), we incorporate the Sinkhorn distance (Cuturi, 2013) and maximum mean discrepancy (MMD; Gretton et al., 2012) for synthetic targets in continuous spaces. Note that EUBO, MMD, and Sinkhorn distance calculations require unbiased samples from the target distribution. For all metrics, we use $K_{\text{eval}} = 2000$ samples.

**ELBO.** The ELBO is defined as:

$$\text{ELBO} = \mathbb{E}_{(\mathbf{x}_{0:N-1}, \mathbf{x}_N = \mathbf{x}) \sim \overrightarrow{p}_\theta} \left[ \log \frac{R(\mathbf{x}) \overleftarrow{p}(\mathbf{x}_{0:N-1}|\mathbf{x})}{\overrightarrow{p}_\theta(\mathbf{x}_{0:N})} \right].$$

The ELBO is a lower bound on the true log-partition function $\log Z$. We use a Monte Carlo estimate using $K_{\text{eval}}$ samples from $\overrightarrow{p}_\theta$.

**EUBO.** The EUBO is defined as

$$\text{EUBO} = \mathbb{E}_{\mathbf{x} \sim \pi, \mathbf{x}_{0:N-1} \sim \overleftarrow{p}(\cdot|\mathbf{x})} \left[ \log \frac{R(\mathbf{x}) \overleftarrow{p}(\mathbf{x}_{0:N-1} \mid \mathbf{x})}{\overrightarrow{p}_\theta(\mathbf{x}_{0:N})} \right]$$

and is an upper bound on $\log Z$. We again use a Monte Carlo estimate with $K_{\text{eval}}$ samples.

**Sinkhorn distance.** We benchmark the Sinkhorn distance for the diffusion sampling task with synthetic targets §4.1. The Sinkhorn distance is the entropic optimal transport cost, using the squared-Euclidean distance and a regularisation parameter of 1, between two batches of $K_{\text{eval}}$ samples: one from the ground truth and one from a trained sampler. We use JAX `ott` library (Cuturi et al., 2022) to compute the Sinkhorn distance, following (Chen et al., 2025).

**MMD.** We compute the MMD between batches of $K_{\text{eval}}$ samples from the ground truth and from the trained sampler. Following Blessing et al. (2024) (§A.2), we use an exponential kernel with a heuristically determined scaling parameter.

### F.3. A Note on Evaluation Protocol

In §4, we report the moving average of each evaluation metric at the end of training, averaged across 5 random seeds. This differs from that of Chen et al. (2025), who report the *best* performance observed during the training process. We argue that our evaluation is more appropriate because, under our assumption of unavailable target samples, metrics such as EUBO, MMD, and the Sinkhorn distance cannot be monitored.

### F.4. Models and Hyperparameters

**Diffusion processes and Parameterisation.** For the diffusion process and neural network architecture, we mostly follow Vargas et al. (2023). First, we consider an Ornstein-Uhlenbeck (OU) process as $\overleftarrow{p}$ and approximate its time-reversal using $\overrightarrow{p}_\theta$. After discretisation, the OU process and its time-reversal with learnable function $f_\theta$ can be written as:

$$\mathbf{y}_{t-\Delta t} = \sqrt{1-\alpha_{1-t}}\mathbf{y}_t + \sigma\sqrt{\alpha_{1-t}}\xi, \quad \xi \sim \mathcal{N}(0,I), \tag{23}$$

$$\mathbf{y}_{t+\Delta t} = \sqrt{1-\alpha_{1-t}}\mathbf{y}_t + 2\sigma^2(1-\sqrt{1-\alpha_{1-t}})f_\theta(\mathbf{y}_t,t) + \sigma\sqrt{\alpha_{1-t}}\xi, \quad \xi \sim \mathcal{N}(0,I), \tag{24}$$

where $\alpha_t = 1 - \exp(-2b_t\Delta t)$ and $b_t$ is non-decreasing positive function with $\int_0^1 b_s ds \gg 1$. In practice, we define $f_\theta(\mathbf{y}_t,t) = \frac{\alpha_{1-t}}{2\sigma^2(1-\sqrt{1-\alpha_{1-t}})}\tilde{f}_\theta(\mathbf{y}_t,t)$ and learn $\tilde{f}_\theta$. See (Vargas et al., 2023) for full derivation.

Let $\Delta t = 1/N$ and denote $\mathbf{x}_n = \mathbf{y}_{n\Delta t}$. Then, we can write $\overleftarrow{p}$ and $\overrightarrow{p}_\theta$ as follows:

$$\overleftarrow{p}(\mathbf{x}_{n-1} \mid \mathbf{x}_n) = \mathcal{N}(\mathbf{x}_{n-1}; \sqrt{1-\alpha_{1-n\Delta t}}\mathbf{x}_n, \sigma^2\alpha_{1-n\Delta t}), \forall n = 1, \dots, N, \tag{25}$$

$$\overrightarrow{p}_\theta(\mathbf{x}_{n+1} \mid \mathbf{x}_n) = \mathcal{N}(\mathbf{x}_{n+1}; \sqrt{1-\alpha_{1-n\Delta t}}\mathbf{x}_n + \alpha_{1-t}\tilde{f}_\theta(\mathbf{x}_n, n\Delta t), \sigma^2\alpha_{1-n\Delta t}), \forall n = 0, \dots, N-1. \tag{26}$$

The learnable function $\tilde{f}_\theta$ is parameterised by a two-layer Multi-Layer Perceptron (MLP). For the gradient-based setting, we use the Langevin parameterisation (LP; (21)), which gives the same architecture as (Vargas et al., 2023). For the gradient-free setting, we remove the LP term, as done in (Sendera et al., 2024; He et al., 2025). We set the number of hidden units ($H_\theta$) of the MLP to 64 when using LP and to 256 otherwise, to compensate for the lack of powerful gradient guidance.

When using SubTB, we use the partial energy scheme introduced in §E to learn the flow functions. The correction term $\tilde{F}_n^\phi$ is also a two-layer MLP, where we set the number of hidden units ($H_\phi$) to 64 for the gradient-free setting and 256 for the gradient-based setting.

**Hyperparameters.** The Table 5 shows our settings for key hyperparameters. We found that our algorithm is robust enough to hyperparameter settings, and thus we did not conduct an extensive hyperparameter search. For TB or TB/SubTB baselines, we use the same hyperparameters as presented in the table, and for other baselines, we follow Blessing et al. (2024) and Chen et al. (2025).

*Table 5.* Hyperparameter settings. See Algorithm 4 for the meaning of $N_{\text{epoch}}, I, K, N, L, \kappa$, and $\gamma$. $\max(|\mathcal{B}|)$ is buffer capacity. $H_\theta$ and $H_\phi$ mean hidden units for $\tilde{f}_\theta$ in (26) and $\tilde{F}_n^\phi$ in (22), respectively. LR means learning rate.

| | Gradient-free | | | | Gradient-based | | | | |
|---|---|---|---|---|---|---|---|---|---|
| Targets → | GMM40 ($d$=2) | GMM40 ($d$=5) | Funnel ($d$=10) | ManyWell ($d$=32) | Funnel ($d$=10) | Robot4 ($d$=10) | GMM40 ($d$=50) | MoS ($d$=50) | ManyWell ($d$=64) |
| $N_{\text{epoch}}$ | | 20000 | | | | | 40000 | | |
| $I$ | | 2 | | | | | 2 | | |
| $K$ | | 2000 | | | | | 2000 | | |
| $N$ | | 64 | | | | | 128 | | |
| $L$ | | 4 | | | | | 4 | | |
| $\kappa$ | | 0.2 | | | | | 0.2 | | |
| $\gamma$ | | 0.05 | | | | | 0.05 | | |
| $\max(|\mathcal{B}|)$ | | 200000 | | | | | 200000 | | |
| $H_\theta$ | | 256 | | | | | 64 | | |
| $H_\phi$ | | 64 | | | | | 256 | | |
| LR for $Z_\theta$ | | $10^{-1}$ | | | | | $10^{-1}$ | | |
| LR for $\beta^\phi$ | | $10^{-1}$ | | | | | $10^{-1}$ | | |
| LR for $\overrightarrow{p}_\theta$ | | $10^{-3}$ | | | $10^{-3}$ | $10^{-3}$ | $10^{-3}$ | $10^{-4}$ | $10^{-3}$ |
| LR for $F^\phi$ | | $10^{-3}$ | | | $10^{-3}$ | $10^{-3}$ | $10^{-3}$ | $10^{-4}$ | $10^{-3}$ |
| $\sigma$ in (26) | 20 | 20 | 1 | 1 | 1 | 2 | 40 | 15 | 1 |

# G. Further Experimental Results

## G.1. Visualisation of Generated Samples

In this section, we visualise generated samples, all of which are taken from the first seed run.

### G.1.1. GRADIENT-FREE SETTING

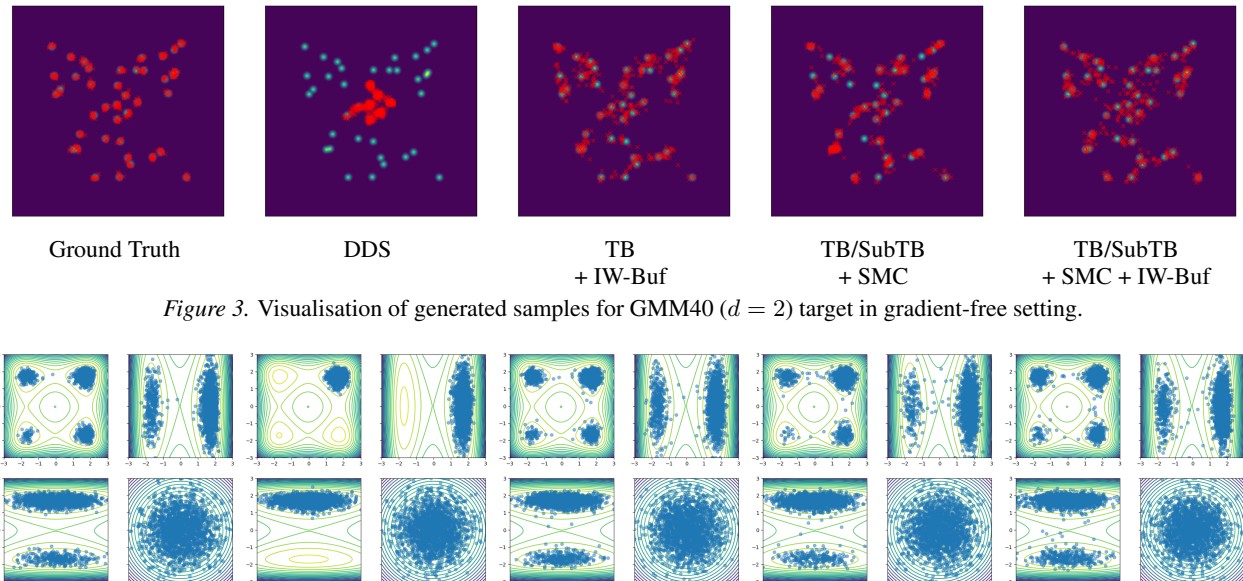

| Ground Truth | DDS | TB
+ IW-Buf | TB/SubTB
+ SMC | TB/SubTB
+ SMC + IW-Buf |

*Figure 3.* Visualisation of generated samples for GMM40 ($d = 2$) target in gradient-free setting.

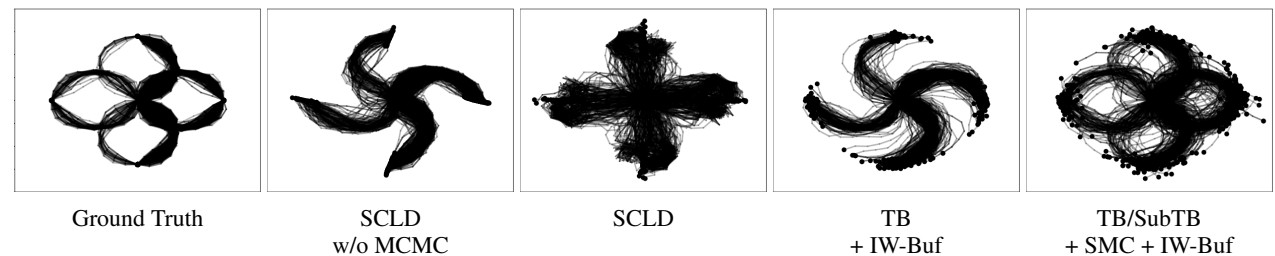

| Ground Truth | DDS | TB
+ IW-Buf | TB/SubTB
+ SMC | TB/SubTB
+ SMC + IW-Buf |

*Figure 4.* Visualisation of generated samples for ManyWell ($d = 32$) target in gradient-free setting.

### G.1.2. GRADIENT-BASED SETTING

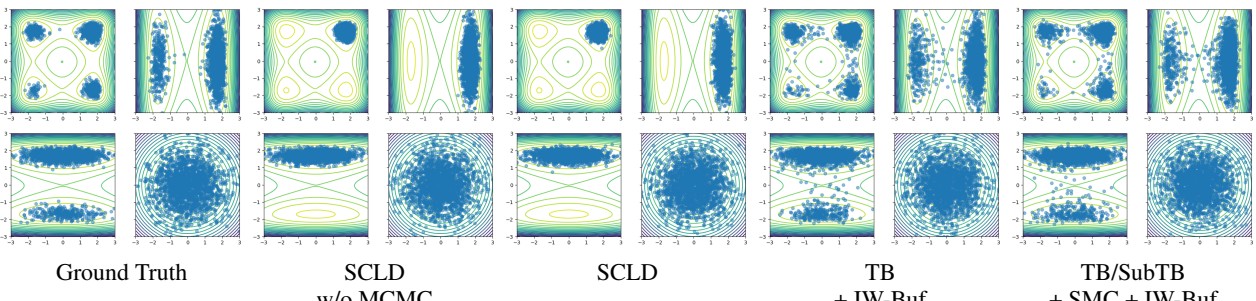

| Ground Truth | SCLD
w/o MCMC | SCLD | TB
+ IW-Buf | TB/SubTB
+ SMC + IW-Buf |

*Figure 5.* Visualisation of generated samples for Robot4 ($d = 10$) target in gradient-based setting. We adopt the visualisation method in (Chen et al., 2025).

| Ground Truth | SCLD
w/o MCMC | SCLD | TB
+ IW-Buf | TB/SubTB
+ SMC + IW-Buf |

*Figure 6.* Visualisation of generated samples for ManyWell ($d = 64$) target in gradient-based setting.

## G.2. Analysis of the Training Loss

In this work, we propose (a) a chunk-based SubTB (20) loss, and (b) mixed usage of TB and SubTB where using (8) for policy parameters $\theta$ and SubTB for the flow parameters $\phi$ (see §3.1 and Algorithm 4). We study both design choices by replacing (a) with widely-used SubTB-$\lambda$ and ablating (b) by using SubTB for both $\theta$ and $\phi$. We use 2d, 5d GMM40 and 32d ManyWell of the gradient-free setting. Both SMC and IW-Buf are employed for this experiment.

*Table 6.* Ablation study on training losses.

| Target $\rightarrow$ | GMM40 ($d = 2$) | | GMM40 ($d = 5$) | | ManyWell ($d = 32$) | |
|---|---|---|---|---|---|---|
| Loss $\downarrow$  Metric $\rightarrow$ | ELBO $\uparrow$ | EUBO $\downarrow$ | ELBO $\uparrow$ | EUBO $\downarrow$ | ELBO $\uparrow$ | EUBO $\downarrow$ |
| **SubTB for both $\theta$ and $\phi$** | | | | | | |
| w/ SubTB-$\lambda$ ($\lambda = 0.5$) | -3.64$_{\pm0.32}$ | 1.10$_{\pm0.05}$ | -31.38$_{\pm3.33}$ | 3.14$_{\pm0.09}$ | 135.58$_{\pm0.31}$ | 175.09$_{\pm0.12}$ |
| w/ SubTB-$\lambda$ ($\lambda = 0.8$) | -4.66$_{\pm0.19}$ | 1.22$_{\pm0.02}$ | -134.44$_{\pm28.58}$ | 5.20$_{\pm0.52}$ | 128.00$_{\pm1.40}$ | 176.09$_{\pm0.20}$ |
| w/ SubTB-$\lambda$ ($\lambda = 0.9$) | -5.13$_{\pm0.39}$ | 1.27$_{\pm0.05}$ | -232.47$_{\pm31.53}$ | 6.85$_{\pm0.65}$ | 128.64$_{\pm1.62}$ | 175.54$_{\pm0.23}$ |
| w/ SubTB-$\lambda$ ($\lambda = 1.0$) | -6.19$_{\pm0.45}$ | 1.39$_{\pm0.04}$ | -316.22$_{\pm11.29}$ | 7.47$_{\pm0.10}$ | 141.93$_{\pm0.50}$ | 176.43$_{\pm0.24}$ |
| w/ SubTB-$\lambda$ ($\lambda = 1.2$) | -12.97$_{\pm1.23}$ | 1.81$_{\pm0.05}$ | -159.51$_{\pm14.40}$ | 10.46$_{\pm4.86}$ | 152.24$_{\pm0.59}$ | 170.53$_{\pm0.20}$ |
| w/ SubTB-$\lambda$ ($\lambda = 1.5$) | -11.34$_{\pm0.95}$ | 1.69$_{\pm0.04}$ | -47.58$_{\pm30.88}$ | 121.54$_{\pm65.80}$ | 157.73$_{\pm2.24}$ | 168.17$_{\pm0.58}$ |
| w/ SubTB-$\lambda$ ($\lambda = 2.0$) | -9.15$_{\pm1.08}$ | 1.53$_{\pm0.07}$ | -14.25$_{\pm5.94}$ | 314.32$_{\pm94.52}$ | 159.71$_{\pm2.52}$ | 167.54$_{\pm0.67}$ |
| w/ SubTB-Chunk ($L = 4$) | -3.09$_{\pm0.28}$ | 1.00$_{\pm0.04}$ | -50.05$_{\pm4.61}$ | 3.14$_{\pm0.07}$ | 133.03$_{\pm0.49}$ | 174.20$_{\pm0.17}$ |
| **TB for $\theta$, SubTB for $\phi$** | | | | | | |
| w/ SubTB-$\lambda$ ($\lambda = 0.5$) | -3.08$_{\pm0.15}$ | 1.02$_{\pm0.02}$ | -16.35$_{\pm0.50}$ | 17.13$_{\pm0.48}$ | 162.50$_{\pm0.17}$ | 166.46$_{\pm0.09}$ |
| w/ SubTB-$\lambda$ ($\lambda = 0.8$) | -3.17$_{\pm0.10}$ | 1.04$_{\pm0.01}$ | -21.34$_{\pm3.70}$ | 5.78$_{\pm3.03}$ | 162.52$_{\pm0.05}$ | 166.43$_{\pm0.03}$ |
| w/ SubTB-$\lambda$ ($\lambda = 0.9$) | -3.10$_{\pm0.13}$ | 1.03$_{\pm0.02}$ | -20.63$_{\pm1.01}$ | 10.60$_{\pm0.81}$ | 162.28$_{\pm0.11}$ | 166.49$_{\pm0.05}$ |
| w/ SubTB-$\lambda$ ($\lambda = 1.0$) | -3.06$_{\pm0.06}$ | 1.02$_{\pm0.01}$ | -30.53$_{\pm2.40}$ | 73.84$_{\pm34.07}$ | 161.73$_{\pm0.16}$ | 166.69$_{\pm0.05}$ |
| w/ SubTB-$\lambda$ ($\lambda = 1.2$) | -3.50$_{\pm0.27}$ | 1.06$_{\pm0.03}$ | -30.17$_{\pm13.04}$ | 224.72$_{\pm223.94}$ | 159.25$_{\pm1.61}$ | 167.51$_{\pm0.42}$ |
| w/ SubTB-$\lambda$ ($\lambda = 1.5$) | -6.20$_{\pm0.38}$ | 1.31$_{\pm0.03}$ | -27.20$_{\pm16.50}$ | 120.82$_{\pm60.69}$ | 159.17$_{\pm2.07}$ | 167.77$_{\pm0.72}$ |
| w/ SubTB-$\lambda$ ($\lambda = 2.0$) | -6.54$_{\pm0.66}$ | 1.33$_{\pm0.06}$ | -26.76$_{\pm10.55}$ | 122.32$_{\pm45.86}$ | 158.00$_{\pm2.66}$ | 173.94$_{\pm11.69}$ |
| w/ SubTB-Chunk ($L = 4$, Ours) | **-2.44$_{\pm0.10}$** | **0.89$_{\pm0.03}$** | **-11.46$_{\pm0.59}$** | **2.3$_{\pm0.1}$** | **162.81$_{\pm0.03}$** | **166.30$_{\pm0.01}$** |

Table 6 shows the results. We can observe that our approach (using both (a) and (b)) achieves the best performance in all cases. The mixed objective (b) improves ELBO in almost all settings. We hypothesise this is because, with (b), the policy parameter $\theta$ is not exposed to the bootstrapping errors of the imperfectly learnt flows. We also note that SubTB-$\lambda$ is sensitive to its hyperparameter $\lambda$, making it difficult to tune in practice.

## G.3. Effect of Adaptive Importance Weight Tempering

In this subsection, we analyse our adaptive importance weight tempering (detailed in §3.4 and Algorithm 5) by varying its threshold, $\gamma$. This threshold defines two extremes: $\gamma = 0$ disables tempering, corresponding to (re)sampling with the original weights, while $\gamma = 1$ makes all weights equal, corresponding to (re)sampling from a uniform distribution.

Here and in other experiments, we use a single $\gamma$ for both SMC and the importance-weighted experience replay; see Algorithm 4 to check where the adaptive tempering occurs. Tuning $\gamma$ independently for each of them could potentially improve performance further.

The results are presented in Fig. 7. High values of $\gamma$ flatten importance weights, which dilutes the effect of importance weighting and degrades performance. Values lower than 0.5 generally give good results, while we empirically found that too small $\gamma$ (e.g., 0.0 or 0.01) can lead to mode collapse or even training instability, particularly for higher-dimensional targets. We use $\gamma = 0.05$ in all other experiments.

## G.4. Hyperparameter Sensitivity

**Subtrajectory length $L$.** The subtrajectory length $L$ defines the number of resampling during SMC sampling (Algorithm 7), since we resample every $L$ steps. If $L = 1$, we attempt resampling at every step, and if $L = N$, we ablate out the SMC, reducing the "TB/SubTB + SMC + IW-Buf" to "TB + IW-Buf". As shown in Fig. 8, smaller $L$ usually gives better results, but it can be costly when we use the partial energy technique (§E) that requires evaluation of $R$ every $L$ step. We use $L = 4$ in §4.

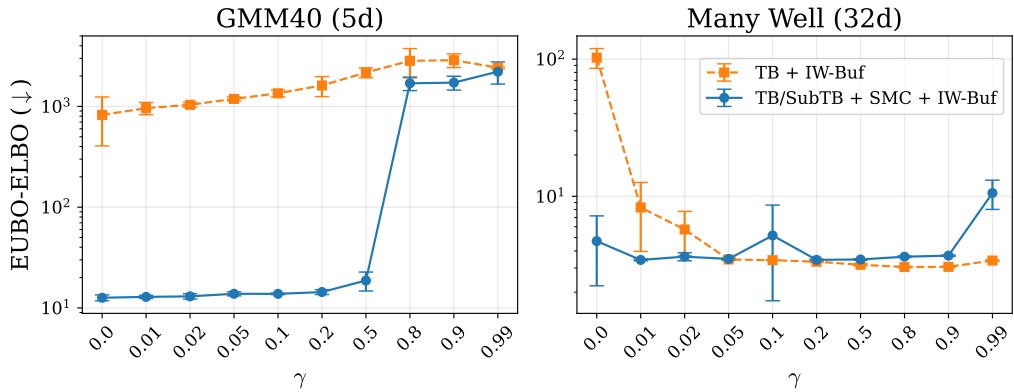

*Figure 7.* Effect of $\gamma$ of adaptive importance weight tempering. Here and in other figures, the error bars show the standard deviation from 5 runs.

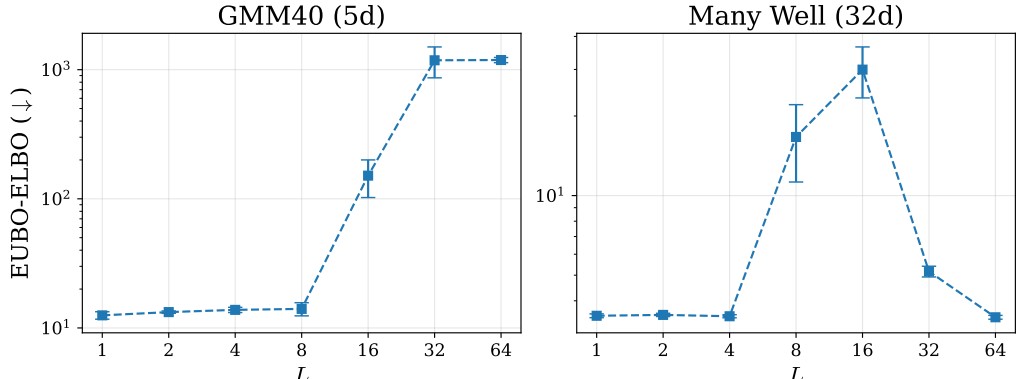

*Figure 8.* Results obtained from different values of subtrajectory length $L$.

**Off-policy ratio $I$.** The off-policy ratio $I$ (see Algorithm 3 or Algorithm 4) specifies the number of off-policy training epochs per on-policy epoch. As shown in Fig. 9, a high ratio ($I \geq 5$) degrades the performance and can lead to unstable training. Note that $I = 0$ corresponds to the on-policy training. We use $I = 2$ for our main experiments in §4.

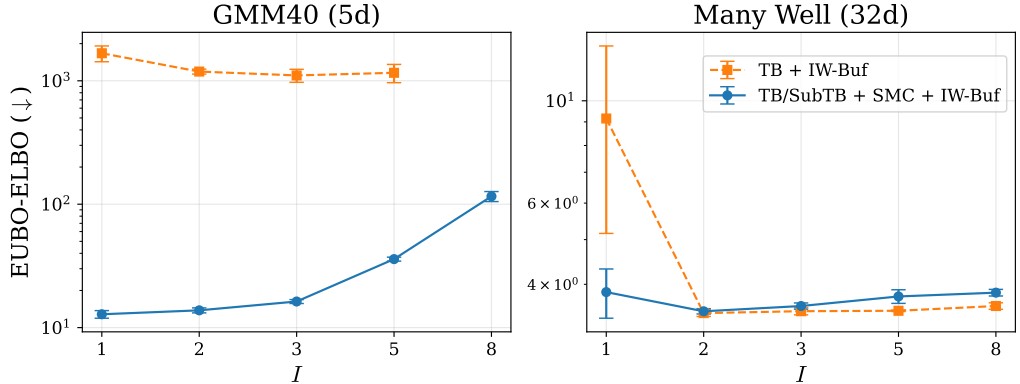

*Figure 9.* Results obtained from different values of replay ratio $I$. Extremely large (above $10^5$) values are omitted.

# H. Application to Alanine Dipeptide

We evaluate our methods for approximating the Boltzmann distribution of alanine dipeptide (ALDP), a 22-atom molecule in an implicit solvent at a temperature of 300K. This challenging molecular sampling task is one of the standard benchmarks for Boltzmann generators (Noé et al., 2019; Wu et al., 2020). We mostly follow the experimental setup of Midgley et al. (2023).

**Target distribution.** Alanine dipeptide is a 22-atom molecule widely used as a benchmark for molecular conformation sampling. The target distribution is the Boltzmann distribution at temperature $T = 300$K:

$$\pi(\mathbf{x}) \propto R(\mathbf{x}) = \exp\left(-\frac{U(\mathbf{x})}{k_B T}\right), \tag{27}$$

where $U(\mathbf{x})$ is the potential energy, $k_B$ is the Boltzmann constant, and $\mathbf{x}$ represents the molecular conformation. Following Midgley et al. (2023), we compute the energy using OpenMM (Eastman et al., 2023). Since this distribution is intractable to sample from directly, we use the validation samples provided by Midgley et al. (2023), which are obtained from long parallel tempering MD simulations.

**Coordinate system.** Following Midgley et al. (2023), we use internal coordinates (bond lengths, bond angles, and dihedral angles) instead of Cartesian coordinates to ensure translational and rotational invariance (Noé et al., 2019; Midgley et al., 2023). The molecule has 60 internal coordinates in total. The backbone dihedral angles $\phi$ and $\psi$ are of particular interest as they characterise the major conformational states of the molecule. Moreover, to focus on the L-form samples, we assign sufficiently high energy to the D-form samples to suppress their generation, unlike Midgley et al. (2023), which filtered the D-form samples after they were generated.

**Model architecture and hyperparameters.** We use the same diffusion process as the synthetic tasks with $\sigma = 1.0$ (See §F.4.) The model architecture was the same as with synthetic tasks, but the MLP hidden dimension was increased to 1,024. We do not use the Langevin parameterisation (§E) due to instability. We set $N_{\text{epochs}}$ 30,000 for algorithms without SMC, and 10,000 for algorithms with SMC, considering the increased number of function evaluations by the partial energy technique (§E). We decrease the batch size to $K = 400$, while we increase the buffer size to 400,000. We use learning rates 0.0005 for the policy $\overrightarrow{p}_\theta$ and flows $F^\phi$, and 0.05 for $Z_\theta$ and $\beta^\phi$. Other hyperparameters remain the same as the synthetic target tasks in §4.

**Buffer augmentation with MCMC.** While our proposed training schemes showed improvement, results were not fully satisfactory because we didn't use gradient (force) information, unlike prior works (Zhang & Chen, 2022; Midgley et al., 2023). To further enhance performance in this challenging task, we augment the buffer with samples from a gradient-informed MCMC algorithm. This approach aligns with the local search methods introduced by Sendera et al. (2024) and further developed in Kim et al. (2025c).

Specifically, we employ underdamped Langevin dynamics tailored for molecular conformation, following Kim et al. (2025c), equipped with BAOAB integration (Leimkuhler & Matthews, 2013). We run multiple independent MCMC chains starting from samples in the buffer; thus, the quality of MCMC samples depends on the buffer type. Every 500 epochs, we run 100 MCMC chains for 500 steps each. We intentionally use a small number of iterations to ensure that performance gains are not solely due to MCMC.

*Table 7.* Results of distribution-based metrics on molecular conformer generation in alanine dipeptide. "×" indicates that training failed due to numerical error.

| Algorithm ↓ Metric → | ELBO (↑) | EUBO (↓) |
|---|---|---|
| PIS | × | × |
| TB | × | × |
| + Buf | × | × |
| + Buf + MCMC | $-755.9_{\pm 268.3}$ | $-125.8_{\pm 1.4}$ |
| + IW-Buf | $-4504.2_{\pm 6740.7}$ | $311.2_{\pm 300.5}$ |
| + IW-Buf + MCMC | $-797.7_{\pm 441.2}$ | $-149.4_{\pm 5.7}$ |
| TB/SubTB | | |
| + SMC + IW-Buf | $\underline{-180.3}_{\pm 1.9}$ | $\mathbf{-166.}_{\pm \mathbf{0.8}}$ |
| + SMC + IW-Buf + MCMC | $\mathbf{-179.5}_{\pm \mathbf{2.7}}$ | $\underline{-155.9}_{\pm 3.5}$ |

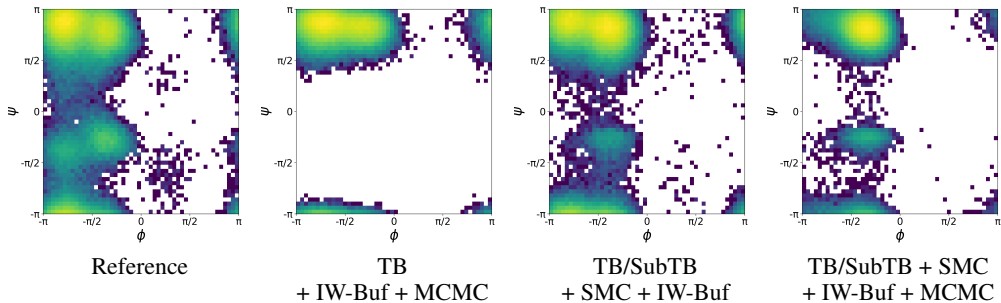

| Reference | TB
+ IW-Buf + MCMC | TB/SubTB
+ SMC + IW-Buf | TB/SubTB + SMC
+ IW-Buf + MCMC |

*Figure 10.* Ramachandran plots generated with 100,000 samples drawn from samplers trained with each algorithm for the first random seed. The reference samples are taken from (Midgley et al., 2023).

## H.1. Results

Table 7 summarises the distribution metrics, and Fig. 10 shows the Ramachandran plots drawn with samples from each model. Due to the highly peaky energy landscape, algorithms without the buffer failed during the optimisation process. The proposed importance-weighted experience replay and SMC scheme significantly improves the results. Still, substantial room for improvement remains, such as using advanced network architectures, finer discretisation steps, incorporating other learning objectives such as likelihood maximisation, or employing more powerful MCMC algorithms.

## I. Application to Biochemical Sequence Design

We follow the experimental settings from Shen et al. (2023) for QM9, sEH, and TFbind8 tasks and from Kim et al. (2024b) for the L14-RNA1 task. For each task, we are given a reward function $R : \mathcal{X} \to \mathbb{R}$ that measures the desirability of a sequence. The target distribution is given by: $\pi(\mathbf{x}) \propto \text{Softplus}(R(\mathbf{x}))^C$, where $\text{Softplus}(x) = \log(1 + \exp(x))$ ensures positivity and $C$ is an exponent factor.

**Target distributions.** We consider four target distributions from past work:

- **QM9** ($|\mathcal{X}| = 58{,}765$). We generate string representations of small molecular graphs with 5 blocks, each sequentially added by prepending or appending from a predefined set of 12 building blocks with 2 stems. The reward is the HOMO-LUMO gap, which we approximate using a pretrained MXMNet model (Zhang et al., 2020). We use $C = 5$.
- **sEH** ($|\mathcal{X}| = 34{,}012{,}224$). Similar to QM9, we create string representations of small molecular graphs with 6 blocks, each from a predefined set of 18 building blocks with 2 stems. The reward is the binding affinity to soluble epoxide hydrolase (sEH), approximated using a pretrained model from Bengio et al. (2021). We set $C = 20$.
- **TFbind8** ($|\mathcal{X}| = 65{,}536$). We generate DNA sequences of length 8, where each token is a DNA nucleotide (A, G, C, or T). The reward function is the binding affinity to a human transcription factor, approximated via a proxy model from (Trabucco et al., 2022). We use $C = 20$.
- **L14-RNA1** ($|\mathcal{X}| = 268{,}435{,}456$). We generate RNA sequences of length 14, where each token is an RNA nucleotide (A, G, C, or U). The reward function is the binding affinity to a human transcription factor, approximated using a model from (Sinai et al., 2020). We set $C = 40$.

**Model architecture and hyperparameters.** We implement prepend/append models (in §B.2) with edge-flow parameterisation following Shen et al. (2023). We use MLP as a backbone architecture, with a hidden dimension of 1,024 for chemical tasks and 128 for biological tasks. We use the Adam (Kingma & Ba, 2015) optimiser with a learning rate of 0.0001 for policy $\overrightarrow{p}_\theta$, and SGD with a learning rate of 0.1 and a momentum coefficient of 0.8 for $\log Z_\theta$. We do not apply learning rate scheduling. We train for 3,000 epochs on QM9 and 6,000 epochs on other tasks, with a batch size of $K = 100$ and a buffer size of 100,000. Other algorithm-specific hyperparameters follow the same settings as the synthetic continuous target tasks.

**Pearson correlation $r$.** We benchmark the sample Pearson correlation coefficient between $\log \overrightarrow{p}_\theta(\mathbf{x})$ and $\log R(\mathbf{x})$ for biochemical sequence design tasks. It is computed with a holdout dataset that follows the target distribution. $r$ ranges from 0 to 1, with 1 indicating a perfect match to the target. Recall from (1), however, $\overrightarrow{p}_\theta(\mathbf{x}) = \int \overrightarrow{p}_\theta(\mathbf{x}_{0:N-1}, \mathbf{x}_N = \mathbf{x}) d\mathbf{x}_{0:N-1}$ is generally intractable to integrate. In discrete space, $\overrightarrow{p}_\theta(\mathbf{x}) = \sum_{\mathbf{x}_{0:N}: \mathbf{x}_N = \mathbf{x}} \overrightarrow{p}_\theta(\mathbf{x}_{0:N})$ could be enumerated in small-scale problems, but it becomes too expensive to enumerate all the possible trajectories for larger problems that we are usually

interested in. Thus, we use the Monte Carlo approximation:

$$\overrightarrow{p}_\theta(\mathbf{x}) = \int \frac{\overleftarrow{p}(\mathbf{x}_{0:N-1} \mid \mathbf{x}_N = \mathbf{x})}{\overleftarrow{p}(\mathbf{x}_{0:N-1} \mid \mathbf{x}_N = \mathbf{x})} \overrightarrow{p}_\theta(\mathbf{x}_{0:N-1}, \mathbf{x}_N = \mathbf{x}) \mathrm{d}\mathbf{x}_{0:N-1}$$

$$= \mathbb{E}_{\mathbf{x}_{0:N-1} \sim \overleftarrow{p}(\cdot|\mathbf{x})} \left[ \frac{\overrightarrow{p}_\theta(\mathbf{x}_{0:N})}{\overleftarrow{p}(\mathbf{x}_{0:N-1}|\mathbf{x})} \right]$$

$$\approx \frac{1}{K} \sum_{i=1}^{K} \frac{\overrightarrow{p}_\theta(\mathbf{x}_{0:N}^i)}{\overleftarrow{p}(\mathbf{x}_{0:N-1}^i|\mathbf{x}^i)}, \quad \mathbf{x}_{0:N-1}^i \sim \overleftarrow{p}(\cdot|\mathbf{x}).$$

We use a one-sample Monte Carlo approximation, *i.e.*, for a given $\mathbf{x}$, we sample a backward trajectory $\mathbf{x}_{0:N}$ using $\overleftarrow{p}$, and then $\overrightarrow{p}_\theta(\mathbf{x}) \approx \frac{\overrightarrow{p}_\theta(\mathbf{x}_{0:N})}{\overleftarrow{p}(\mathbf{x}_{0:N-1}|\mathbf{x})}$, *i.e.*, $\log \overrightarrow{p}_\theta(\mathbf{x}) \approx \log \overrightarrow{p}_\theta(\mathbf{x}_{0:N}) - \log \overleftarrow{p}(\mathbf{x}_{0:N-1}|\mathbf{x})$. The estimate has zero variance if TB (8) is minimised for all trajectories ending in $\mathbf{x}$.

*Table 8.* Results on chemical sequence design tasks.

| Target → | QM9 | | | sEH | | |
|---|---|---|---|---|---|---|
| Algorithm ↓ Metric → | ELBO (↑) | EUBO (↓) | $r$ (↑) | ELBO (↑) | EUBO (↓) | $r$ (↑) |
| TB | **21.591**$_{\pm0.002}$ | 21.605$_{\pm0.001}$ | **0.951**$_{\pm0.003}$ | 52.502$_{\pm0.041}$ | 54.082$_{\pm0.475}$ | 0.629$_{\pm0.089}$ |
| + $\epsilon$-expl. | 21.590$_{\pm0.002}$ | 21.606$_{\pm0.001}$ | 0.948$_{\pm0.002}$ | 52.527$_{\pm0.021}$ | 53.291$_{\pm0.098}$ | **0.894**$_{\pm0.011}$ |
| + Buf | 21.590$_{\pm0.001}$ | 21.607$_{\pm0.002}$ | 0.946$_{\pm0.003}$ | 52.500$_{\pm0.017}$ | 53.249$_{\pm0.066}$ | 0.881$_{\pm0.009}$ |
| + R-Buf | 21.587$_{\pm0.002}$ | 21.607$_{\pm0.002}$ | 0.944$_{\pm0.003}$ | 52.475$_{\pm0.020}$ | 53.312$_{\pm0.316}$ | 0.827$_{\pm0.075}$ |
| + L-Buf | 21.589$_{\pm0.001}$ | 21.606$_{\pm0.002}$ | 0.943$_{\pm0.002}$ | 52.510$_{\pm0.016}$ | 53.250$_{\pm0.019}$ | 0.880$_{\pm0.007}$ |
| + IW-Buf (ours) | 21.590$_{\pm0.001}$ | **21.604**$_{\pm0.001}$ | 0.946$_{\pm0.002}$ | **52.537**$_{\pm0.038}$ | **52.844**$_{\pm0.015}$ | 0.863$_{\pm0.004}$ |

*Table 9.* Results on biological sequence design tasks.

| Target → | TFbind8 | | | L14-RNA1 | | |
|---|---|---|---|---|---|---|
| Algorithm ↓ Metric → | ELBO (↑) | EUBO (↓) | $r$ (↑) | ELBO (↑) | EUBO (↓) | $r$ (↑) |
| TB | **12.272**$_{\pm0.018}$ | 13.086$_{\pm0.028}$ | 0.810$_{\pm0.007}$ | 17.181$_{\pm0.038}$ | 19.955$_{\pm0.024}$ | 0.785$_{\pm0.001}$ |
| + $\epsilon$-expl. | 12.271$_{\pm0.014}$ | 13.075$_{\pm0.016}$ | 0.806$_{\pm0.005}$ | 17.098$_{\pm0.036}$ | 20.047$_{\pm0.040}$ | 0.781$_{\pm0.003}$ |
| + Buf | 12.258$_{\pm0.016}$ | 13.069$_{\pm0.015}$ | 0.804$_{\pm0.006}$ | 17.101$_{\pm0.021}$ | 20.027$_{\pm0.025}$ | 0.782$_{\pm0.002}$ |
| + R-Buf | 12.247$_{\pm0.014}$ | 13.096$_{\pm0.024}$ | 0.818$_{\pm0.007}$ | **17.349**$_{\pm0.025}$ | 19.970$_{\pm0.025}$ | 0.794$_{\pm0.003}$ |
| + L-Buf | 12.271$_{\pm0.011}$ | 13.058$_{\pm0.015}$ | 0.810$_{\pm0.005}$ | 17.179$_{\pm0.049}$ | 19.941$_{\pm0.025}$ | 0.790$_{\pm0.002}$ |
| + IW-Buf (ours) | 12.209$_{\pm0.023}$ | **12.977**$_{\pm0.012}$ | **0.843**$_{\pm0.006}$ | 17.225$_{\pm0.026}$ | **19.764**$_{\pm0.024}$ | **0.827**$_{\pm0.002}$ |

## I.1. Results

We compare our proposed importance-weighted experience replay techniques, IW-Buf, against other off-policy training schemes, including $\epsilon$-exploration (Bengio et al., 2021), and various buffer prioritisation schemes. Tables 8 and 9 show the results. We can observe that IW-Buf improves mode coverage, as indicated by consistently lower EUBO. Additionally, our methods yield comparable or slightly superior results to baselines in other metrics. Note that these results were achieved without changing key hyperparameters related to the importance of weighting, meaning that our methods are readily applicable to a broad class of amortised sampling problems in both continuous space and discrete space.

