# OpenReview forum: "Reinforced Sequential Monte Carlo for Amortised Sampling"
_ICML.cc/2026/Conference — ICML 2026 spotlight_

### Official Review · Reviewer_55ez · 2026-02-28

**Soundness:** 3
**Presentation:** 3
**Significance:** 3
**Originality:** 3
**Overall Recommendation:** 5
**Confidence:** 3

**Summary:**

The goal is to sample from an unnormalized target distribution $R(x)$ on a continuous or discrete space. To do so, the authors alternate between two steps:

**Step 1: Sequential Monte Carlo (SMC)**:  run a Markov Chain between each pair in a sequence of distributions $p_0, \ldots, p_N := R(x) / Z$ that approach the target.

**Step 2: Tune the design choices to improve the SMC: intermediate distributions, noising Markov Kernels, denoising Markov Kernels**: here the noising Markov kernels $p_B$ are kept fixed. The two other design choices are tuned by minimizing a cost function (table below, row 12). This cost function is computed using samples from *any* trajectories: in practice, the authors sample from a mixture of noising trajectories (off-policy, do not depend on $\theta$) and denoising trajectories (on-policy, do depend on $\theta$). This cost function has many equivalent interpretations: matching the noising and denoising trajectories (table below, row 11), making the noising and denoising kernels $p_F$ and $p_B$ Bayes-inverses, making the reweighting step in SMC unecessary (Eq 9), or maximizing certain rewards and the stochasticity of the policy from a Reinforcement Learning perspective (table below, row 11).

Experimental results are encouraging.

| # | Sequential Monte Carlo | Reinforcement Learning |
|---|-----------------------------------|------------------------|
|   | **Vocabulary** |                        |
| 1 | iterate $x_n$ | state $s_n$ |
| 2 | next iterate $x_{n+1}$ | action $a_n$ |
| 3 | trajectory $\tau = (x_1, \ldots, x_N)$ | trajectory (reformulated) $\tau = ((s_1, a_1), \ldots, (s_N, T))$ |
|   | **Probabilistic model** |                        |
| 4 | denoising Markov kernel $p_F(x_{n+1} \mid x_n; \theta)$ | policy model $p_F(a_n \mid s_n; \theta)$ |
| 5 | denoising Markov kernel (reformulated) $p_F(x_{n+1} \mid x_n, x_{n+1})$ | world dynamics $p_F(s_{n+1} \mid s_n, a_n)$ |
| 6 | noising Markov kernel $p_B(x_n \mid x_{n+1})$ | intermediate reward $r(s_n, a_n)$ |
| 7 | joint denoising distribution $p_F(\tau; \theta) = p_0(x_0)\prod_{n=0}^{N-1} p_F(x_{n+1} \mid x_n; \theta)$ |  |
| 8 | joint noising distribution $p_B(\tau; \theta) = R(x_N)\prod_{n=0}^{N-1} p_B(x_n \mid x_{n+1}) / Z$ |  |
|   | **Objective function** |                        |
| 9 | log unnormalized target $\log R(x_N)$ | terminal reward $r(s_N, \mathrm{terminate})$ |
| 10 | log unnormalized intermediate densities $\log F_n^{\psi}(x)$ | soft value functions $V_n^{\text{soft}}(s)$ |
| 11 | KL between trajectory laws $\mathrm{KL}(p_F(\tau; \theta) \| p_B(\tau; \theta)) \geq \mathrm{KL}(p(x_N; \theta) \| \pi(x_N))$ | Maximum-Entropy RL $\mathbb{E}_{\tau \sim p_F(\tau; \theta)}[ \sum r(s_n, a_n) + \mathcal{H}(p_F(.  \mid x_n; \theta)) ]$ |
| 12 | Sub-trajectory loss (enforces Bellman) $\mathcal{L}^{\theta,\phi}(\tau_{m:n}) = \left[ \log \frac{F_m^{\phi}(x_m)\,\prod p_F(x_{i+1} \mid x_i; \theta)}{F_n^{\phi}(x_n)\,\prod p_B(x_i \mid x_{i+1})} \right]^2$ |  |

**Compliance With Llm Reviewing Policy:**

Affirmed.

**Final Justification:**

The authors have answered my questions. Scaling sampling algorithms that have only access to the unnormalized density is an important topic and this work seems rather comprehensive.

**Key Questions For Authors:**

Q1. What choice of noising kernel $\overleftarrow{p}$ do you use in the experiments? This choice seems crucial given that the cost you minimize encourages the sampling policy to be its Bayes-inverse.

Q2. In table 3, is there a reason why certain metrics (MMD and Sinkhorn) are used for certain datasets (GMM40), and other metrics (ELBO, EUBO) are used for other datasets (ManyWell)?

**Limitations:**

Yes

**Strengths And Weaknesses:**

**Strengths**: the paper presentation is clear. The experiments are encouraging and provide an empirical answer to the question of "what are good design choices (intermediate distributions and denoising kernels) for the SMC algorithm?". Table 3 and Appendix G2 clearly show that the geometric interpolation (Eq 11) does *not* provide good intermediate distributions for SMC. This concurs with the conclusion of recent theoretical works that could be worth referencing: Eq 18 of [1] and Section 4.1 of [2]. Appendix G2 also shows that the learnt intermediate densities are *transport* mass smoothly from the reference distribution to the target distribution. The authors also investigate many tricks (Section 3) to make the optimization work in practice.

**Weaknesses**:  the experiments remain low-dimensional but the authors acknowledge this. Section 3, and in particular section 3.4., outlines a "bag of tricks" to make the experiments work in practice. It is not clear to me how brittle the method is if so many tricks are necessary.

[1] Mate et al. Learning Interpolations between Boltzmann Densities. TMLR, 2024.

[2] Chehab et al. Provable Convergence and Limitations of Geometric Tempering for Langevin Dynamics. ICLR 2025.

---

> ### Author Rebuttal · Authors · 2026-03-30
>
> Thank you for your detailed review and constructive feedback.
>
> ## Theory on geometric annealing for intermediate targets (in Strengths)
>
> Thank you for these suggestions.
>
> To recall, the intermediate marginals $F_n^\phi$ in our paper are defined as geometric annealing with a learnable correction or 'twist' term: $F_n^{\phi}(x_n) = \overrightarrow{p}_0(x_n)^{1-\beta_n} R(x_n)^{\beta_n} \tilde{F}_n^{\phi}(x_n)$ (eq. (26)).
>
> We are aware of the papers [1,2] you suggest. We already cite [1] in Appendices A, E, and F (and will mention it more prominently in the main text). The geometric interpolation there indeed motivates our choice of intermediate density, and we cite recent work [Berner et al., "From discrete-time policies..."] that shows that the subtrajectory-based objectives studied in this paper converge in the continuous-time limit to the PDEs that [1] optimises directly. We will emphasize this connection more clearly in the revised manuscript.
>
> Regarding [2], our results are consistent with the fact that geometric annealing is suboptimal (see §G.2 and Table 3). Indeed, learning the intermediate targets as corrections to the geometric annealing is beneficial.
>
> ## Sensitivity to parameters and stabilisation techniques
>
> While an effective integration of SMC and off-policy RL training requires some algorithm choices, we respectfully disagree with the characterisation of §3.4  as a "bag of tricks". We make a number of choices that are *already standard in diffusion-based sampling* (see §E). Similarly, resampling is standard in SMC; our contribution is in how we couple adaptive resampling/tempering with off-policy RL training and IW replay in this amortised diffusion sampling setting.
>
> Please see the response to Reviewer wZMR ("Is the approach sensitive to hyperparameters?"), where we point to the existing ablation studies (§4.3 and §G.3-§G.5) and give some additional results on sensitivity to the key hyperparameters.
>
> ## Choice of $\overleftarrow p$
>
> The choice of noising/backward kernel is a standard design choice in diffusion-based sampling. We consider a time-discretised Ornstein-Uhlenbeck (OU) process in the main experiments (see §F.4).
>
> We show below the results of an additional experiment using pinned Brownian motion, which is also common in the diffusion samplers literature, instead, keeping all other training choices the same. The results are shown in Table A. We observe that PBM performs worse than the OU process in the gradient-free setting, but our proposed off-policy mechanisms (SMC and IW-Buf) still yield large gains over the baselines.
>
> **Table A: Sinkhorn distance (mean±std over 5 runs) results with different $\overleftarrow{p}$ in the gradient-free setting.**
>
> |Algorithm|GMM40 ($d=2$)|GMM40 ($d=5$)|ManyWell ($d=32$)|
> |:--|:--:|:--:|:--:|
> |**OU**||||
> |DDS|595.54±9.94|3009.6±0.8|29.58±0.02|
> |TB|607.31±12.05|3110.2±121.4|29.57±0.02|
> |TB + IW-Buf|6.50±0.11|2813.9±133.1|22.97±0.02|
> |TB/SubTB + SMC|39.99±9.49|330.9±185.0|21.91±0.06|
> |TB/SubTB + SMC + IW-Buf|6.46±0.40|83.3±10.1|22.97±0.04|
> |**PBM**|||
> |PIS|600.26±7.01|3003.3±114.8|29.48±0.02|
> |TB|614.57±2.29|3032.7±139.6|29.49±0.02|
> |TB + IW-Buf|8.70±0.73|2922.2±237.8|23.18±0.06|
> |TB/SubTB + SMC|579.22±14.93|diverged|22.63±0.07|
> |TB/SubTB + SMC + IW-Buf|7.64±0.85|125.5±80.0|23.13±0.07|
>
> ## Metrics in Table 3
>
> We simply chose metrics to be consistent with Tables 1 and 2 for each dataset (given the space limit); we are happy to report all of the metrics. See Tables B and C below ($\times$ indicates severe training instability). Please also refer to L381 (left) in the paper for the selection of evaluation metrics.
>
> **Table B**
> |ManyWell($d=32$)|MMD↓|Sinkhorn↓|
> |--|:--:|:--:|
> |Learnt $F^{\phi}_n$ (No interp.)|0.039±0.014|23.02±0.11|
> |Linear interp.|0.051±0.017|23.15±0.12|
> |+Learnt $\tilde{F}^{\phi}_n$|0.046±0.012|23.11±0.08|
> |Cosine interp.|0.045±0.015|23.19±0.11|
> |+Learnt $\tilde{F}^{\phi}_n$|0.038±0.011|23.10±0.06|
> |Learnt interp.|0.052±0.022|22.97±0.17|
> |+Learnt $\tilde{F}^{\phi}_n$(Ours)|0.033±0.001|22.97±0.04|
>
> **Table C**
> |GMM40($d=50$)|ELBO↑|ELBO↓|
> |--|:--:|:--:|
> |Learnt $F^{\phi}_n$ (No interp.)|-11.72±0.07|10.67±0.02|
> |Linear interp.|$\times$|$\times$|
> |+Learnt $\tilde{F}^{\phi}_n$|-11.87±0.07|10.77±0.01|
> |Cosine interp.|$\times$|$\times$|
> |+Learnt $\tilde{F}^{\phi}_n$|-11.87±0.07|10.78±0.01|
> |Learnt interp.|-11.62±0.08|10.61±0.02|
> |+Learnt $\tilde{F}^{\phi}_n$(Ours)|-11.50±0.15|10.52±0.01|

---

> > ### Author Rebuttal · Reviewer_55ez · 2026-04-04
> >
> > Thanks authors. All my questions have been answered, I am happy to update my score and recommend acceptance.

---

### Official Review · Reviewer_tZrQ · 2026-03-09

**Soundness:** 4
**Presentation:** 4
**Significance:** 3
**Originality:** 4
**Overall Recommendation:** 5
**Confidence:** 4

**Summary:**

The authors provide a unifying framework for heirarchical variational inference scheme and maximum entropy reinforcement learning and use this framework to combine the benefits of amortized samplers and Monte Carlo methods. This paper proposes to use using neural components to generalize and perform sampling in finite number of steps and training the sampler with adaptive tempering and experience replay using SMC samples for  off-policy training. The proposed approach works for both continuous and discrete spaces and multi-modal distributions.

**Compliance With Llm Reviewing Policy:**

Affirmed.

**Key Questions For Authors:**

1- How do the authors compare their work to adjoint sampling algorithm where  the number of target energy function evaluations is minimized leading to efficient multistep sampling approach? Is it possible to benefit from a similar property with off-policy training approach?

**Limitations:**

yes.

**Strengths And Weaknesses:**

Strengths: The proposed approach in the paper is grounded. The authors reformulate sampling problem in terms of Reinforcement learning objective and leverage from existing developments in this area for optimization and implementation choices.

Presentation: The paper is well-written and structured. The motivation, problem formulation and method are carefully explained and accompanied by detailed background, and reasoned design and implementation procedure. Experiments include challenging problems that are used for benchmarking in state-of the art amortized sampling approaches.

Significance: Sampling from unnormalized probability distributions is one of the important challenges in multiple scientific areas and the paper proposes a method to improve the existing amortized samplers by incorporating methods from RL for better exploration and optimization during training.

Novelty: The approach uses techniques in Reinforcement learning to train hierarchical neural samplers and combines the advantages of both methods, leading to better coverage of the multi-modal target distributions in high dimensional spaces.

---

> ### Author Rebuttal · Authors · 2026-03-30
>
> Thank you for your thoughtful review and positive assessment.
>
> ## Adjoint sampling
>
> Adjoint sampling methods require a single target evaluation for each training trajectory, as does our method.
>
> However, they require access to the gradient of the target distribution to update the estimated score function at intermediate time steps, while our method does not -- it assumes a black-box target. In addition, our method is fully off-policy, allowing the use of a replay buffer populated using SMC to construct the behaviour policy, while adjoint sampling methods are semi-on-policy (requiring an on-policy rollout of the forward kernel beginning at a given intermediate state to obtain a regression target).
>
> An interesting approach for future exploration would be to select intermediate states from which to perform on-policy rollouts using SMC-based buffer techniques such as those we study in this paper.

---

> > ### Author Rebuttal · Reviewer_tZrQ · 2026-04-03
> >
> > I thank the authors for their answer to my question. The comments address my concern about method's efficiency and improve my understanding of the paper.

---

### Official Review · Reviewer_wZMR · 2026-03-11

**Soundness:** 3
**Presentation:** 3
**Significance:** 3
**Originality:** 3
**Overall Recommendation:** 4
**Confidence:** 2

**Summary:**

The paper proposes a fusion of amortized and particle-based sampling method for sampling from unnormalized densities (i.e. without access to samples).  The paper’s core idea is: it connects hierarchical variational inference, maximum-entropy RL, and SMC/AIS in one framework, then uses that connection in both directions — the learned sampler supplies SMC proposals and twisted/intermediate targets, while SMC-generated samples become off-policy training data for the sampler. The paper explores techniques for stable joint training (via variance reduction) and replay buffers. Empirically the manuscript demonstrates on synthetic multimodal tasks in both continuous and discrete spaces, as well as on Boltzmann distributions for alanine dipeptide conformations, improvements compared to previous methods.

**Compliance With Llm Reviewing Policy:**

Affirmed.

**Final Justification:**

The authors have fully addressed my concerns and I do recommend acceptance. The RL side doesnt really fall into my area of expertise (hence I will keep my scores). I do also agree with the authors comment, that a unified framework of  RL/Deep Learning and classical sampling approaches is very useful and relevant.

**Key Questions For Authors:**

- How sensitive is performance to the main hyperparameters? Were hyperparameters selected per-task (to which degree?)
- How much extra computation does the full method require?

**Limitations:**

Yes the paper discusses limitations in "future work" as well as in the appendix.

**Strengths And Weaknesses:**

Strength:
- The main idea is interesting and well-motivated: it builds a bridge between amortized sampling, maximum-entropy RL, and SMC in a single framework. The paper is very well written and method is clearly presented.
- Quite broad empirical analysis which show gains in most (but not all) task that were considered (stronger in some metrics, weaker in others).

Weaknesses:
- SMC baselines: As I understood the method is more comparable to SMC with inner kernel tunings i.e. a sampler that adopts the transition kernels parameters on the fly (or in the proposed case, learns the kernel). This can be some simple schemes like estimating the mass matrix from particles + step-size tuning based on previous acceptance rates, or more elaborate schemes (e.g. Fearnhead, P., & Taylor, B. M. (2013). An adaptive sequential Monte Carlo sampler or Buchholz, A., Chopin, N., & Jacob, P. E. (2021). Adaptive tuning of hamiltonian monte carlo within sequential monte carlo. Bayesian Analysis, 16(3), 745-771.).
- The overall approach seems quite complex with many hyperparameters to which the performance is sensitive to.

---

> ### Author Rebuttal · Authors · 2026-03-30
>
> Thank you for your good questions and constructive feedback.
>
> ## SMC baselines
>
> Thank you for the suggested references. These are about adapting the transition kernels at runtime on the level of a single SMC run, which is different from our approach of learning a fixed amortised sampler that does not require any runtime adaptation, with the kernels adapted (learnt) over the course of an RL training loop in which involves SMC runs.
>
> To clarify the goal, the main object we are hoping to obtain is a generative model sampler that can produce samples from the target distribution with a single fixed-time rollout, without the need for runtime particle-based methods; indeed, we evaluate the resulting models without resampling. The novelty of our work is in the way we use SMC to facilitate off-policy training of this amortised sampler. Please also see the response to Reviewer Cxex.
>
> **We also take this chance to comment on the significance and originality of our work in the context of the SMC literature.** Broadly, there have been two approaches to hierarchical sampling that have evolved somewhat independently: using SMC with adaptive (possibly learnt) intermediate targets and kernels (mainly in the statistics literature) and using fully learnt generative models for sampling with no runtime SMC or MCMC, typically trained by RL-like algorithms (mainly in the machine learning literature).
>
> Our work places these two approaches in a common framework by building a dictionary between RL, hierarchical VI and SMC, which has not been functionally done before, although many connections among these have been noted and used before. In our paper, this dictionary allows a transfer of ideas and algorithms between these views on the sampling problem: SMC informs the design of an off-policy RL algorithm and the use of importance weights for experience replay; RL algorithms are used for the learning of SMC kernels and intermediate targets. End-to-end RL for learning of twist functions has not been done before to the best of our knowledge, and the resulting algorithms successfully outperform baselines.
>
> ## Is the approach sensitive to hyperparameters?
>
> First, we kindly remind you that our paper already includes hyperparameter setup (Table 5 in §F.4) as well as ablation studies on key algorithmic design choices in §4.3 and §G.3-5. We have not done extensive hyperparameter tuning and set key hyperparameters the same across targets for each experiment.
>
> Here we provide additional analysis on the adaptive resampling threshold $\kappa$ with **TB/SubTB + SMC + IW-Buf** algorithm. The lower the $\kappa$ is, the less the resampling is used. The results are presented in the Table A below, showing that our algorithm is robust to change of $\kappa$. We observed similar stability in the absence of IW-Buf (i.e., TB/SubTB + SMC) for $\kappa \geq 0.1$.
>
> **Table A: Sinkhorn distance (mean±std over 5 runs) for different $\kappa$.**
>
> |Algorithm|GMM40 ($d=5$)|ManyWell ($d=32$)|GMM40 ($d=50$)|
> |:--|:--:|:--:|:--:|
> |$\kappa=0.01$|79.13±9.38|22.99±0.01|3770.33±183.80|
> |$\kappa=0.02$|77.67±7.33|22.96±0.07|3717.43±276.72|
> |$\kappa=0.05$|83.22±8.34|22.87±0.11|3713.84±245.40|
> |$\kappa=0.1$|82.35±11.59|22.87±0.05|3630.00±270.00|
> |$\kappa=0.2$ (default)|83.31±10.12|22.97±0.04|3597.17±163.43|
> |$\kappa=0.5$|75.85±9.32|22.87±0.06|3573.55±273.06|
> |$\kappa=1.0$|82.42±8.23|22.90±0.06|3632.61±261.80|
>
> ## How much extra computation?
>
> We can measure computation cost using the number of target function evaluations (NFE). Simply put, SMC variants result in only a moderate increase in NFE in the gradient-based setting, while it requires much more NFE in the gradient-free setting compared to baselines (due to partial energy scheme, §E). We adjust SMC variants to make their NFE comparable to baselines, and report results in Tables B and C. While the adjusted SMC variants become worse than before, they still give improvement over on-policy training.
>
> **Table B: Gradient-free setting.** Sinkhorn distance (mean±std over 5 runs) and NFE are reported.
> |Algorithm|GMM40 ($d=5$)|ManyWell ($d=32$)|NFE ($\times 10^7$)|
> |:--|:--:|:--:|:--:|
> |DDS|3009.6 ± 0.8|29.58 ± 0.02|4.00|
> |TB|3110.2 ± 121.4|29.57 ± 0.02|4.00|
> |TB/SubTB + SMC|330.9 ± 185.0|21.91 ± 0.06|104.00|
> |TB/SubTB + SMC (**adjusted**)|2027.8 ± 287.1|29.10 ± 0.37|3.95|
> |TB/SubTB + SMC + IW-Buf|83.3 ± 10.1|22.97 ± 0.04|104.00|
> |TB/SubTB + SMC + IW-Buf (**adjusted**)|354.3 ± 82.2|26.94 ± 0.51|3.95|
>
> **Table C: Gradient-based setting.** Sinkhorn distance (mean±std over 5 runs) and NFE are reported.
> |Algorithm|Robot4 ($d=10$)|GMM40 ($d=50$)|NFE ($\times 10^{10}$)|
> |:--|:--:|:--:|:--:|
> |DDS|$\times$|6882.66±125.25|1.024|
> |SCLD|11.24±4.76|7477.14±489.84|$\approx$ 0.84|
> |TB|1.72±0.01|3903.95±139.18|1.024|
> |TB/SubTB + SMC|64.48±103.61|$\times$|1.707|
> |TB/SubTB + SMC (**adjusted**)|5.23±2.24|$\times$|0.853|
> |TB/SubTB + SMC + IW-Buf|0.39±0.44|3579.17±163.43|1.707|
> |TB/SubTB + SMC + IW-Buf (**adjusted**)|0.58±0.41|3735.56±199.99|0.853|

---

> > ### Author Rebuttal · Reviewer_wZMR · 2026-04-02
> >
> > I thank the authors for the detailed response. This has addressed my main concerns, I have not further questions for the author.

---

### Official Review · Reviewer_Cxex · 2026-03-13

**Soundness:** 4
**Presentation:** 3
**Significance:** 3
**Originality:** 2
**Overall Recommendation:** 5
**Confidence:** 4

**Summary:**

The paper connects reinforcement learning to the choice of (optimal) proposal distributions in a sequential Monte Carlo (SMC) sampler. The authors describe optimal SMC sampler proposals as an instance of a maximum entropy off-policy reinforcement learning objective, using hierarchical variational inference as an underlying model. Previous work has considered variational methods for the underlying Markov model in SMC to learn 'optimal' proposals. Here, reinforcement learning is used in place of other variational methods.

**Compliance With Llm Reviewing Policy:**

Affirmed.

**Final Justification:**

The authors addressed my concerns and clarified some points for which I am not an expert. The paper crosses boundaries between Bayes computation and machine learning in a novel and effective way. I think it is a important contribution to the area.

**Key Questions For Authors:**

1. When discussing the use of normalising constants (NC) to weight batches of particles in the replay buffer (Section 3.3), I didn't understand how the NCs prioritised different batches. Shouldn't the estimated NCs all be the same in expectation? If so, I don't understand the principle behind their use.

2. When discussing obtaining trajectories for training with reverse posterior factorisation, is this related to forward-backward sampling used in particle smoothing? Or ancestor sampling?

3. I found it curious to bias the SMC sampler by tempering the weights to maintain ESS (Section 3.4). Adaptive SMC samplers will choose an annealing schedule adaptively to manage this. Is this because you need a fixed schedule in the sampler (set of intermediate target distributions)? And do you only temper during training, then use untempered version for the final iteration? A version of the proposal learning that can handle an adaptive schedule (size and targets) would be very compelling.

**Limitations:**

Yes.

**Strengths And Weaknesses:**

Soundness: In general the paper is technically sound (see questions for more details) and I found the experiments compelling and discussed sufficiently in the appendices.

I found some use of terminology confusing. In particular, the emphasis on amortisation throughout the paper appears to be a misnomer. I would expect an amortised version of the proposed algorithm to be w.r.t. a unnormalised target distribution $R(x \mid y)$ over a distribution for $y$. However, the target is fixed as $R(x)$. I also found the used of 'twist'/'twisted'/'twisting' to be used incorrectly. I could not locate where a twisted proposal or distribution is used. That is, a proposal $p(x)$ would be twisted if considering $p^\prime(x) \propto p(x) \psi(x)$ for some non-negative integrable function $\psi(x)$. In SMC this has been used in [1,2] and other places, but there could be a difference in terminology that I am missing. The use of 'optimality' (pg 5, ln 222) also needs to be stated carefully. The canonical divergence for SMC algorithms is $\chi^2$ whilst the authors discuss KL, and optimality only occurs when the class of approximating distributions are rich enough.

Presentation: In general, the paper is well-written and structured. Given the developed method is an SMC algorithm, the emphasis on previous reinforcement learning and its literature seemed somewhat high compared to sequential Monte Carlo. There is a rich literature of choosing optimal proposal kernels and intermediate distributions. Important recent works such as [1,2,3] were not considered. The comments on the benefits of objectives that do not require importance weights when 'off-policy', also applies to learning twisting functions in [1,2] for example.

One minor comment: I found it difficult on first reading to understand that the reverse order factorisation of the posterior was not related to the $p_\theta$'s because at that point no (extended) posterior has been introduced.

Small error: should 'release' be 'realise' (pg 8, col 2, ln 399)

Significance: The use of reinforcement learning tools to learn 'optimal' proposals is an interesting and useful idea. I found the integration between SMC and reinforcement learning to be slightly superficial (perhaps due to limited discussion of past SMC research) but also compared to a recent paper in using reinforcement learning for MCMC [4] where the integration is highly novel. This indicates to me that more sophisticated integration could be possible, but this paper is a good starting point.

Originality: The borrowing of loss functions (e.g., subtrajectory balance) from reinforcement learning for SMC proposals is novel, as is experience replay. I found the other claims of contributions (hierarchical variational inference, maximum entropy connection to optimal SMC proposals) to be less compelling in terms of originality.


[1] Guarniero, P, et al. "The iterated auxiliary particle filter." Journal of the American Statistical Association 112.520 (2017): 1636-1647.

[2] Heng, J, et al. "Controlled sequential monte carlo." The Annals of Statistics 48.5 (2020): 2904-2929.

[3] Syed, Saifuddin, et al. "Optimised annealed sequential Monte Carlo samplers." arXiv preprint arXiv:2408.12057 (2024).

[4] Wang, C., Chen, et al. "Reinforcement Learning for Adaptive MCMC." AISTATS 258:640-648 (2025)

---

> ### Author Rebuttal · Authors · 2026-03-30
>
> Thank you for your good questions and constructive feedback.
>
> ## On terminology
> We will clarify the below meanings in the paper.
>
> ### Amortisation
> There are two kinds of 'amortisation' to consider:
> - Amortising the cost of a runtime sampler (SMC/MCMC) into a model that, once trained, requires only one fixed-time rollout to produce samples (the setting we consider).
> - If the target distribution is conditional: amortising across conditions, which is what you mention.
>
> The latter is often meant in "amortised variational inference", but the former is also called "amortised inference/sampling" in the diffusion-based sampling literature. Some of this literature considers *both* kinds simultaneously -- making the sampler conditional (e.g., the VAE encoder in [Sendera et al., "Improved off-policy...", 2024]).
>
> ### Twist
> The term 'twist' as used in [1-3] refers to modification of proposal kernels. However, the term has been used more broadly to mean tilting of intermediate target distributions, as in [4-6] and in our paper. Please also see the response to Reviewer 55ez on our choice of intermediate targets.
>
> ### Optimality
> The statement on L222 is about global optimality in the space of *all* transition kernels, which holds regardless of the divergence used.
>
> While $\chi^2$ is natural to consider when bounding the importance sampling error (due to its direct connection to the standard IS estimator), much of the literature on diffusion-based sampling for posterior estimation targets KL minimisation.
>
> ## Discussion of SMC
> To make our presentation of SMC literature more thorough, we will add to §A a discussion of the works you mentioned, among others (e.g., [4-6]).
>
> Here is a summary of the key differences between *your [1-3]* and our work:
> - *Your [3]* is about optimising the annealing schedule for intermediate distributions given fixed kernels, while our work is about learning both the kernels and the intermediate distributions.
> - *Your [1]* (iAPF) introduced the optimal form of twist for transition kernels for particle filters and an algorithm to approximate it. *Your [2]* (cSMC) extended iAPF to sampling from a static distribution.
> - Our work shares cSMC's idea of fixing the (extended) target distribution using the reverse factorisation and finding the optimal proposal kernels and intermediate distributions.
>   - However, cSMC introduces (generally intractable) integrals in importance weights, which restricts $\overrightarrow{p}$ and $\psi$ to specific parametrised classes. On the other hand, we model $\overrightarrow{p}$ and $F^{\phi}$ independently as neural networks, allowing for much broader function classes.
>   - In terms of learning, as you pointed out, dynamic-programming approaches like iAPF and cSMC also do not require importance weighting to train with historical samples, so our IW buffer can be used.
>
> ## Significance/originality
> Please see the response to Reviewer wZMR.
>
> The integration of SMC, VI, and RL builds a bridge between the two literatures that shows empirical promise and opens directions for future work. We hope these responses, and the additional discussion of SMC we will include, will clarify the novelty and significance.
>
> ## NCs as batch weights
> The idea of using normalising constants as batch-level weights is motivated by group importance sampling [7] and is equivalent to re-normalising all weights. This scheme satisfies Liu's definition of proper weighting. Please refer to Theorem 1 and §B of [7].
>
> ## Reverse sampling related to particle smoothing?
> While both involve sampling trajectories backward from a terminal sample, the goals are different. The goal of particle smoothing is to approximate posterior marginals or joint distributions over intermediate states under a given model. In contrast, the reverse factorisation used for off-policy learning is rather a way to sample trajectories in proportion to their likelihood under the optimal forward sampler conditional on a terminal sample.
>
> ## Weight tempering
> The motivation for weight tempering is to mitigate path degeneracy. For the diffusion sampler, adaptive SMC [8] can be applied to our framework, since the intermediate distributions can be defined in continuous time (i.e., the number of intermediate targets is arbitrary). We agree that combining the adaptive SMC idea with diffusion samplers can be an interesting direction for future work.
>
> We use tempering only for training and evaluate the resulting sampler *without resampling* for fair comparison with baselines.
>
> ---
> ## References
> [1] Whiteley et al. Twisted particle filters, 2014
>
> [2] Guarniero et al. The iterated auxiliary particle filter, 2017
>
> [3] Heng et al. Controlled sequential Monte Carlo, 2020
>
> [4] Lawson et al. Twisted variational sequential Monte Carlo, 2018
>
> [5] Lawson et al. Sixo..., 2022
>
> [6] Zhao et al. Probabilistic inference in language models..., 2024
>
> [7] Martino et al. Group importance sampling..., 2018
>
> [8] Zhou et al. Toward automatic model comparison..., 2016

---

> > ### Author Rebuttal · Reviewer_Cxex · 2026-04-02
> >
> > I thank the authors for a well thought out response. The comments have addressed my concerns and further clarify my understanding of the paper.

---

### Decision · Program_Chairs · 2026-04-30

**Decision:**

Accept (spotlight)

**Comment:**

Clear consensus to accept from all reviewers, with positive comments on overall soundness, presentation, and the overall success of the paper in bringing to use reinforcement learning for developing "optimal" proposals for an SMC algorithm. All reviewers found the paper interesting and I think that this will be a useful paper for helping bridge the semantic and formulation gap between RL and SMC methods which do in fact have much in common.

Please do incorporate all the suggested changes into the final document, particularly the extended discussion of relevant SMC literature that came up in the rebuttal to reviewer Cxex.

There were some concerns from reviewers around potential brittleness in practice, due to the large number of parameters, but this is not a major concern.